# The anti-neural role of BMP signaling is a consequence of its ancestral function in dorsoventral patterning

Paul Knabl [1,2,3], June F. Ordoñez[2,4], Juan Daniel Montenegro Cabrera[1], Daniel Abed-Navandi[5], Roland Halbauer[6], Oliver Link[1,2], Tim Wollesen[4], Grigory Genikhovich[1]*

1 Department of Neurosciences and Developmental Biology, University of Vienna, Vienna, Austria,
2 Vienna Doctoral School of Ecology and Evolution (VDSEE), University of Vienna, Vienna, Austria,
3 Konrad Lorenz Institute for Evolution and Cognition Research, Klosterneuburg, Austria, 4 Department of Evolutionary Biology, University of Vienna, Vienna, Austria, 5 Haus des Meeres, Vienna, Austria,
6 Tiergarten Schönbrunn, Vienna, Austria

* grigory.genikhovich@univie.ac.at

## Abstract

In Bilateria with centralized nervous systems (e.g., in vertebrates or arthropods), the minimum of the BMP signaling activity gradient defines the position of the central nervous system. BMP-dependent patterning of the secondary body axis is ancestral for Bilateria and possibly also for the bilaterian sister clade Cnidaria. However, the variety of levels of centralization of the nervous systems in Bilateria—from diffuse to fully centralized—as well as the lack of centralization of the nervous system in Cnidaria, suggest that BMP signaling cannot be perceived as a universally "anti-neural" signal. Here we use transgenic reporter lines in the anthozoan cnidarian *Nematostella* to show that BMP signaling is active in distinct neuronal populations. Moreover, attenuation of BMP signaling followed by RNA-Seq shows that BMP signaling is a positive regulator of many neuronal genes, including the top-tier neural progenitor marker *soxB(2)*. Furthermore, we analyze BMP signaling activity in the cubozoan jellyfish *Tripedalia* and the scyphozoan jellyfish *Stomolophus* and prove that BMP signaling in parts of the diffuse nervous system is not an anthozoan but an ancestral cnidarian feature, shared by anthozoans and medusozoans. Finally, we show that the highly centralized ventral nervous system of the nonmodel spiralian, the chaetognath *Spadella*, forms out of paired BMP signaling-positive domains on the lateral sides of the embryo. Together, our data suggest that one of the ancestral roles of BMP signaling may have been in promoting neurogenesis. We propose that the "anti-neural" function of BMP signaling in vertebrates and arthropods is a consequence of its global role in the dorsoventral patterning of the ectoderm.

**Data availability statement:** Newly generated sequencing data is deposited as bioprojects at https://www.ncbi.nlm.nih.gov/bioproject/PRJNA1269470 and http://www.ncbi.nlm.nih.gov/bioproject/1441074. All code required to reproduce the computational analyses of this study can be found on https://github.com/Genikhovich-Lab/ and, additionally, at https://doi.org/10.5281/zenodo.19686897 and https://doi.org/10.5281/zenodo.19686882. All other relevant data are within the paper and its Supporting information files.

**Funding:** This research was funded by the Austrian Science Fund (FWF) grants P32705 and PAT2395824 to G.G. and the Austrian Science Fund (FWF) grant P34665 to T.W. During his time in the Genikhovich lab, P.K.'s salary was paid from the grants to G.G. P.K. is a recipient of the Dimitrov Fellowship of the Austrian Academy of Sciences (OeAW) and of the Writing-up Fellowship of the Konrad Lorenz Institute for Evolution and Cognition Research (KLI). During her time in the Wollesen lab, J.O.'s salary was paid from the grant to T.W. The funders had no role in study design, data collection and analysis, decision to publish, or preparation of the manuscript.

**Competing interests:** The authors have declared that no competing interests exist.

**Abbreviations:** bHLH, basic helix-loop-helix; CNS, centralized nervous system; DMH1, Dorsomorphin Homolog 1; DV, dorsoventral; hpf, hours post fertilization; NM, Nematostella Medium; NGS, normal goat serum; PMR, parietal muscle region; SD, Standard deviation; SEM, standard error of the mean.

## Introduction

The evolutionary origin of BMP signaling-dependent regulation of animal nervous systems is uncertain. Both in vertebrates and arthropods, the establishment of a BMP signaling gradient is required for patterning the dorsoventral (DV) body axis and the formation of the centralized nervous system (CNS). In these models, the position of the neuroectoderm is defined by the inhibition of BMP signaling on one side of the DV axis—dorsally in vertebrates and ventrally in arthropods [1]. Later during development, similar genetic programs regulate the mediolateral patterning of the vertebrate neural plate and the nerve cords of insects and annelid worms [2,3]. These observations made in the mouse, frog, zebrafish, fly, and ragworm have contributed to the idea that the CNS emerged once, early during bilaterian evolution on the side of the body, where BMP signaling was low [4–8]. DV axis inversion at the base of chordates made this ancestrally ventral BMP minimum dorsal in chordates [9].

While the role of BMP signaling in regulating the DV patterning is clearly ancestral and, with few exceptions, present across all Bilateria [10], its strictly "anti-neural" function is contradicted by the variety in the degrees of centralization of bilaterian nervous systems—from diffuse, to being arranged into a varying number of nerve cords, to ganglionic, to fully centralized [11,12]. Similarly variable are the effects of BMP signaling perturbations on the nervous systems of different groups, from promoting to suppressing neurogenesis to having no effect at all [13–18]. Clearly, in order to understand the ancestral role of BMP signaling in neuron formation we need a wider phylogenetic sampling within and outside Bilateria, especially in animals with diffuse nervous systems and with centralized nervous systems forming in "nonstandard" locations.

The evolutionary sister clade to Bilateria is the phylum Cnidaria, encompassing Anthozoa (sea anemones and corals) and four classes of Medusozoa: Hydrozoa (hydroids), Staurozoa (stalked jellyfish), Scyphozoa (true jellyfish), and Cubozoa (box jellyfish) (Fig 1A). Cnidaria possess a full repertoire of BMP pathway genes, including BMP ligands, BMP antagonists, BMP receptors and BMP effectors [19–25]. Cnidarian nervous systems are organized as diffuse nerve nets with local condensations, forming nerve rings and neurite tracts, but lacking a brain-like centralization [26–29]). Neurons are found both in the epidermis and in the gastrodermis of cnidarians, and, strikingly, work on the anthozoan sea anemone *Nematostella vectensis* showed that gastrodermal neurons are "born" in the gastrodermis rather than migrate there, as is the case in Bilateria [28]. Neuronal gene expression in *Nematostella* commences already in the blastula, which is before pSMAD1/5 becomes first detectable [30–32]. Neuronal markers were not affected by the disruption of BMP signaling during early phases of neurogenesis [32]. Only at planula larva stage, the upregulation and, surprisingly, the downregulation of BMP signaling both reduced neuronal markers and the number of RFamide- and GLWamide-positive neurons [32] suggesting that neural "induction" is independent of BMP signals, while later neurogenesis may involve a combination of BMP activation and repression. Taken together, it is largely unclear if and to what extent BMP signals are required during cnidarian neurogenesis.

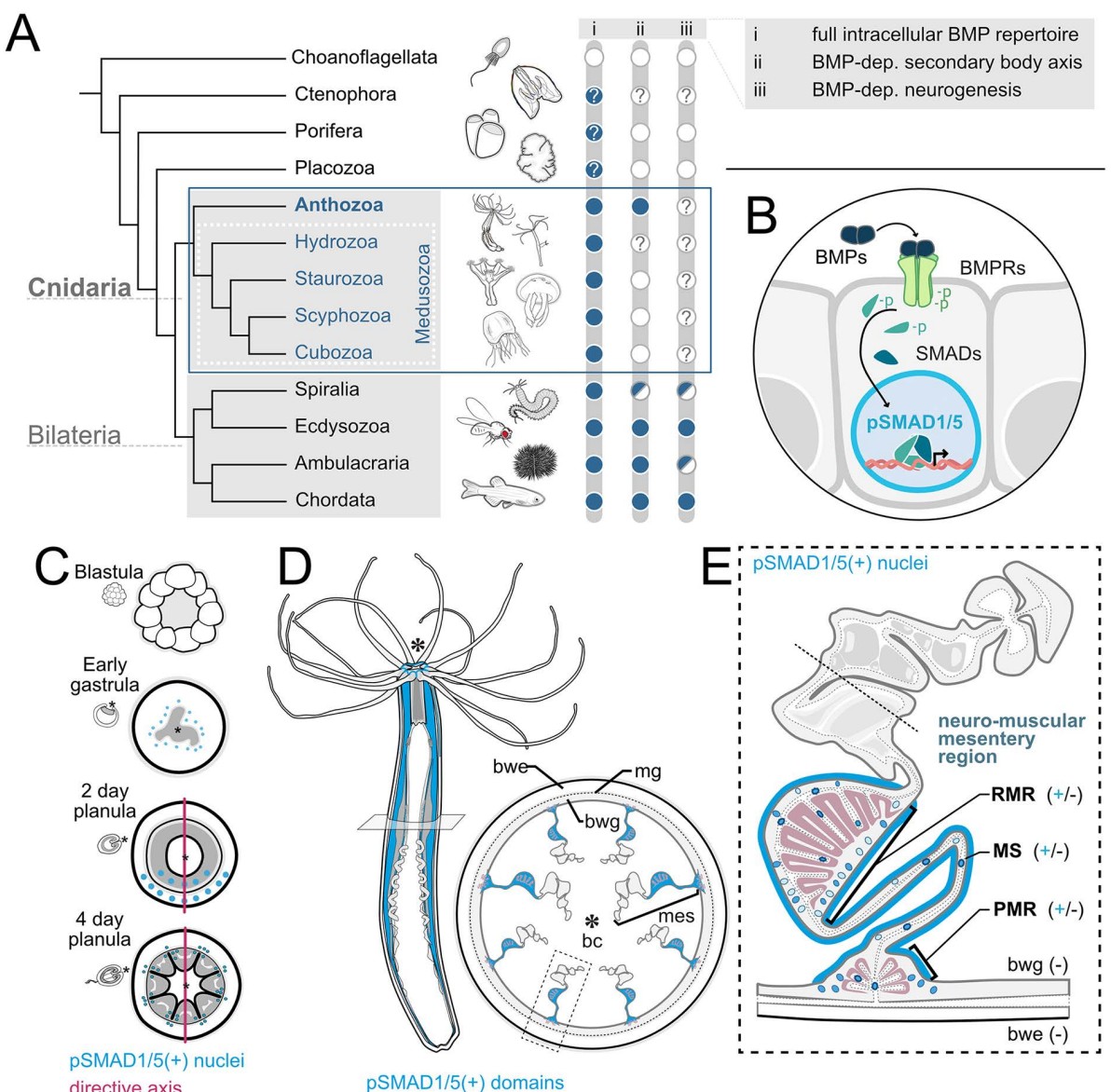

**Fig 1. BMP signaling in *Nematostella vectensis*. (A)** Simplified phylogenetic tree of early branching metazoans indicating (i) the presence or absence of BMP pathway genes, (ii) the presence of a BMP-dependent secondary body axis, (iii) involvement of BMP signaling in neurogenesis. White circle – absent, blue circle – present; half blue/half white circle – present in some; blue and white circles with question marks – possibly present or absent, respectively. **(B)** Schematic of the BMP signaling pathway, BMP signaling is activated by the binding of BMP dimers to the BMP receptor complex, resulting in the nuclear translocation of phosphorylated SMAD1/5 (pSMAD1/5), acting as BMP effector and regulating BMP-responsive gene expression. **(C)** BMP signaling dynamics during early development of *Nematostella*: no activity in the blastula, pSMAD1/5 activity around the blastopore in the early gastrula, pSMAD1/5 gradient at 2-day planula stage and dispersed pSMAD1/5 activity in the 4-day planula. Blue dots indicate nuclear pSMAD1/5. **(D)** Overview of *Nematostella* polyp anatomy in lateral view and as a cross-section of the body column, the neuro-muscular domain of the mesentery gastro-dermis is highlighted in blue. **(E)** Detailed schematic of an individual mesentery as a cross-section, highlighting the localization of pSMAD1/5-positive nuclei within the neuro-muscular domain. bwe-body wall epidermis, mg-mesoglea, bwg-body wall gastrodermis, mes-mesenteries, bc-body cavity, RMR-retractor muscle region, MS-mesentery stalk, PMR-parietal muscle region.

Most of our knowledge about BMP signaling in Cnidaria comes from an anthozoan model—the sea anemone *Nematostella vectensis*. Unlike radially symmetric Medusozoa, *Nematostella* and other Anthozoa, display a bilaterally symmetric body plan with a secondary, "directive" body axis (Fig 1A). Similar to the bilaterian DV axis, the directive axis requires the formation of a BMP signaling gradient that is regulated by the interplay of multiple BMP pathway components, including *bmp2/4, bmp5-8, gdf5-like* (=*gdf5-l*)*, chordin, gremlin*, *rgm* and *zswim4-6* [30,33–35]. Active BMP signaling can be visualized by the spatial distribution of its transcriptional effector, phosphorylated SMAD1/5 (pSMAD1/5). In *Nematostella*, at late gastrula stage, BMP signaling forms a pSMAD1/5 gradient along the directive axis. This gradient persists while the directive axis is being patterned, but eventually disappears by 4 day planula larva stage, at which point BMP signaling activity becomes local and especially prominent in the mesenteries—the gastrodermal septa of the body wall [30,33,34]. Our recent study revealed pronounced BMP signaling activity in the mesenteries of the adult *Nematostella* polyp [36]. The mesenteries harbor a diversity of cell populations including epithelial, digestive, reproductive, muscular and neuronal cells. The two latter are concentrated in the "neuro-muscular" mesentery region, where the longitudinal muscles (the retractor and parietal muscle), the longitudinal neurite tracts, and sensory and ganglion mesentery neurons are located. While BMP signaling is strongest within the neuro-muscular domain, yet it was unclear if BMP signaling occurs in neuronal or muscular cell populations (Fig 1). Here, we analyze the potential involvement of BMP signaling in the diffuse nervous system of the anthozoan cnidarian *Nematostella*. Additionally, we use two other distantly related medusozoan cnidarians, *Tripedalia* and *Stomolophus* for comparison and show a pro-neural action of BMP signaling in these models. Finally, we also demonstrate distinct BMP activity in the central nervous system of the nonmodel spiralian *Spadella*. Together, our findings suggest that BMP signaling was ancestrally a pro-neural factor.

## Results

### BMP pathway genes are expressed across neuronal subtypes in the developmental single-cell atlas of *Nematostella*

To address the expression of BMP components in the neuroglandular and muscular repertoire of *Nematostella*, we utilized the updated developmental single-cell atlas of *Nematostella* [37]. The analysis of the neuroglandular subset by Cole and colleagues yielded 47 developmentally closely related but distinct cell states, including *insulinoma (insm)*-positive (N1) and insm-negative (N2) neurons, secretory cells (S) and digestive gland cells (GD). Among the N1 and N2 neurons, several gastrodermis-derived types have been identified so-far, including six N1 (N1.g1-6) and a single N2 type (N2.g). The analysis of the retractor muscle subset by Cole and colleagues revealed four distinct cell states, including the early retractor muscle state (RM.1), the late tentacle retractor muscle state (TR.2), the early tentacle retractor muscle state (TR.1), and the late retractor muscle state (MR.1). We generated expression plots of BMP pathway genes, including BMP ligands (*bmp2/4, bmp5-8, gdf5-l, admp*), BMP effectors (*smad1/5, smad4, smad4-like*), BMP receptors (*alk2, alk3/6, bmprII, actrII*) and intracellular, membrane-bound, and secreted BMP antagonists (*rgm, smad6, chordin, gremlinA, gremlinB, noggin1, noggin2, crossveinless2 and follistatin*) [36]. The expression of BMP components was detectable in N1 and N2 neuron types, while it was reduced in putative sensory neurons (N1S) and secretory cells (S) and absent in digestive gland (GD) types (Fig 2A). Notably, the expression of BMP components in N1 and N2 neurons is not limited to the gastrodermal populations, despite the fact that our previous analysis revealed that BMP signaling activity is largely limited to the mesentery gastrodermis [36]. The expression of extracellular molecules such as BMP ligands and antagonists appeared to be sparse with only small fractions of cells (<10%) in each subpopulation expressing the transcripts (Fig 2A). Moreover, single-cell analysis suggested that neither BMP pathway genes nor BMP receptor type expression is specific to discrete neuronal subpopulations. While our expression analysis of the developmental single-cell data pointed to the expression of the BMP module in neuronal populations, it was not sufficient to identify specific candidates among the neuronal subpopulations, which might display pSMAD1/5 activity. The expression of BMP pathway genes in the retractor muscle cell states was low or absent, except the

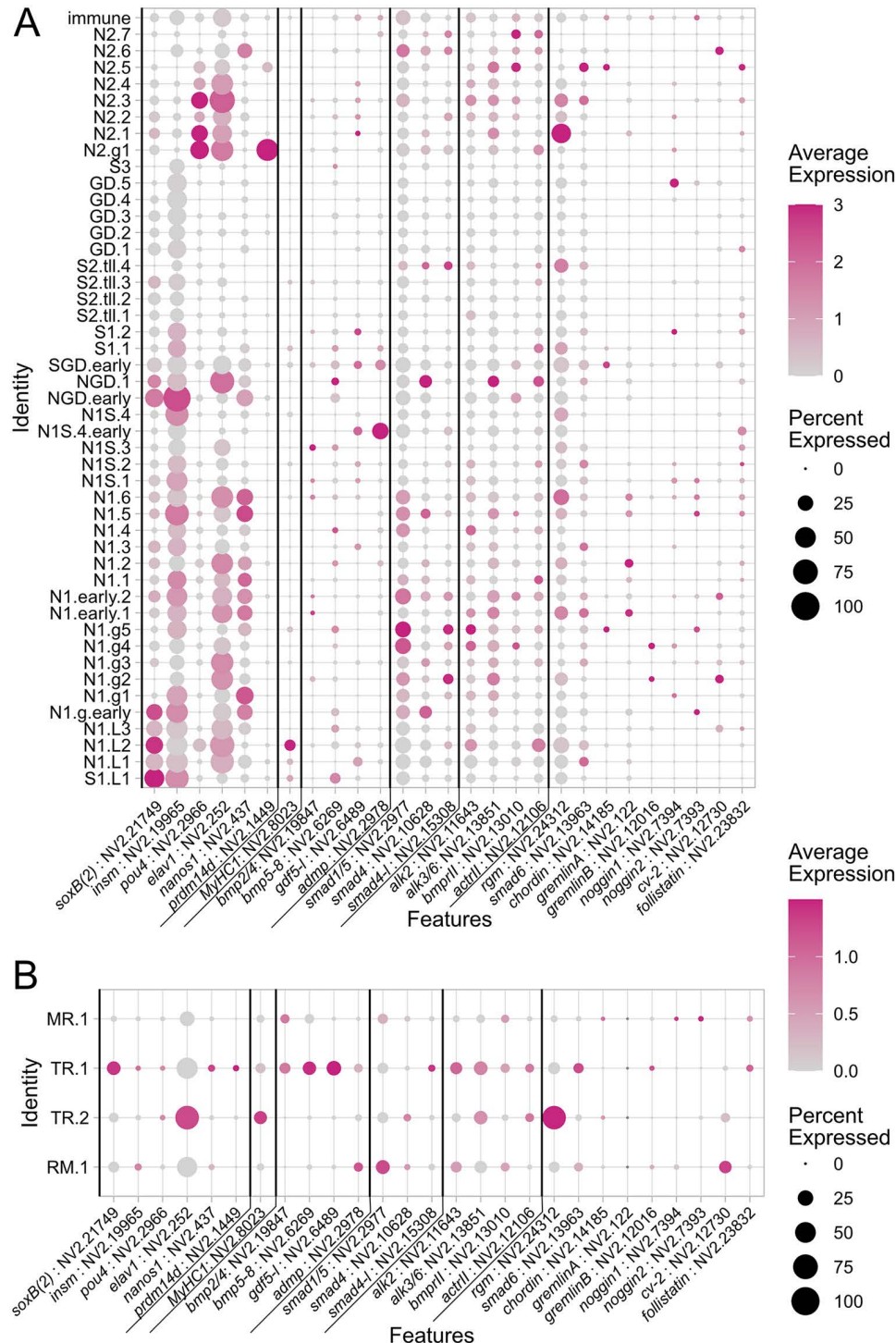

**Fig 2. Expression of BMP pathway genes in the neuronal and retractor muscle subset of the developmental single-cell atlas of *Nematostella*.** Dot plots showing the expression of neuronal marker genes, retractor muscle marker *myosin heavy chain1* (*myhc1*) and BMP signaling components **(A)** in the neuroglandular subset and the **(B)** retractor muscle subset of the *Nematostella* developmental single-cell by Cole *et al*. [37]. Average scaled expression of 0 or below is indicated in gray. Numerical data for the plot can be extracted by running the R scripts found here https://doi.org/10.5281/zenodo.19686897.

upregulated expression of BMP ligands in the early tentacle retractor muscle TR.1 and of the BMP antagonist mole-cule RGM in the late tentacle retractor muscle state TR.2 (Fig 2B).

**pSMAD1/5 is active in the neurons but not in the muscle cells of the neuro-muscular region of the mesentery**

Given the upregulated expression of BMP pathway genes in neuron types according to our single-cell analysis, we aimed to determine if BMP signaling is indeed active in neural cell populations using different transgenic reporter lines. Earlier, we showed that BMP signaling activity in the adult polyp is most pronounced in the neuro-muscular region of the mes-entery gastrodermis [36], therefore, we focused our analysis on reporter lines with expression in gastrodermal neurons and muscles. In the neuro-muscular region, accumulations of different neurons are found along the retractor and the pari-etal muscles as well as in the mesentery stalk (Figs 1E and 3). Firstly, we examined the retractor muscle region, contain-ing the retractor muscle and multiple ganglionic and sensory neuron populations. The expression plot of the single-cell retractor muscle subset suggests low expression levels of BMP components in the retractor muscle (Fig 2B). To determine if BMP signaling is active in the retractor muscle cells, we performed anti-pSMAD1/5 antibody staining in the *myosin heavy chain1::mCherry* (*myhc1::mCh*) transgenic reporter line, expressing mCherry under the *myhc1* promoter in the retractor muscles in the mesenteries as well as in the epidermal muscles of the tentacles [38]. In *myhc1::mCh* trans-genic animals, we observed pSMAD1/5-positive cells occurring within the gastrodermal epithelium or within the mesoglea as basoepithelial cells of the retractor muscle region (Fig 3B, 3C). These pSMAD1/5-positive cells, however, displayed no overlap with mCherry-positive retractor muscle cells (Fig 3C–3F'), suggesting BMP signaling is not active in the retractor muscle itself.

Next, we tested if BMP signaling is active in neuronal subpopulations with known localization in the neuro-muscular mesentery, including *soxB(2)*, *elav1*, *nanos1* and *prdm14d* expressing cells [28,39–43]. The *soxB(2)::mOr* reporter line was previously shown to mark neuronal progenitors and differentiated neurons in the regions of the retractor muscle, pari-etal muscle and body wall gastrodermis [41]. We observed a partial overlap of *soxB(2)* expression and pSMAD1/5 activity in cells of the retractor muscle region, primarily located within the mesoglea (Fig 3G, white arrowheads). These basoepithelial cells were mostly pSMAD1/5- and *soxB(2)*-double-positive, while we rarely observed only pSMAD1/5-positive or only *soxB2*-positive cells, suggesting that a large portion of neural progenitor cells of the mesentery receive BMP signals (Fig 3G–3H'). We then examined transgenic reporter lines for *elav1* and *nanos1*, marking sensory and ganglionic neurons in the gastrodermis and epidermis [28,43] and another *nanos* paralog, *nanos2*, which labels a broad range of cell types in both body layers, including neuroglandular populations in the mesentery [39]. In *elav1::mCherry*, *nanos1::mCherry* and *nanos2::mCherry,* we discerned an overlap of pSMAD1/5 with neuronal populations (Fig 3I–3K", white arrowheads). For *nanos1,* double-positive mCherry and pSMAD1/5 cells had a ganglionic neuron morphology (Fig 3J–3J"), while for *elav1* and *nanos2*, we found double-positive cells, which looked like sensory neurons (Fig 3I–3I' and 3K–3K"). The situation was different when we analyzed the *prdm14d::gfp* reporter line, in which GFP marks ganglionic neurons in the mesentery retractor and parietal muscle region [40]. In the *prdm14d::gfp* line, GFP-positive cells never dis-played BMP signaling activity (Fig 3L–3L"), suggesting BMP signaling in the mesentery retractor region is activated only in distinct neuronal subtypes.

Next, we investigated the basal domain of the neuro-muscular mesentery, where the parietal muscle region is located. Here, pSMAD1/5-positive cells form two gastrodermal clusters that are ambilateral to the parietal muscles and follow the entire primary body axis length (Fig 4). The same area is known to contain the longitudinal neural tracts, two parallel bundles of concentrated neurites on each side of the parietal muscle. To check if BMP signaling coincides with neurons in the parietal muscle region, we analyzed pSMAD1/5 immunoreactivity in the reporter lines for *soxB(2)*, *elav1, nanos1* and *prdm14d,* which have been demonstrated to label the neural tracts [26–28,31,43,44]. First, we tested if BMP sig-naling is active in *soxB2::mOr* cells and detected pSMAD1/5 in mOrange-positive basiepithelial cells in the mesoglea of the mesentery stalk and parietal muscle region (Fig 4B–4C', white arrowheads). In addition, BMP signaling was active in

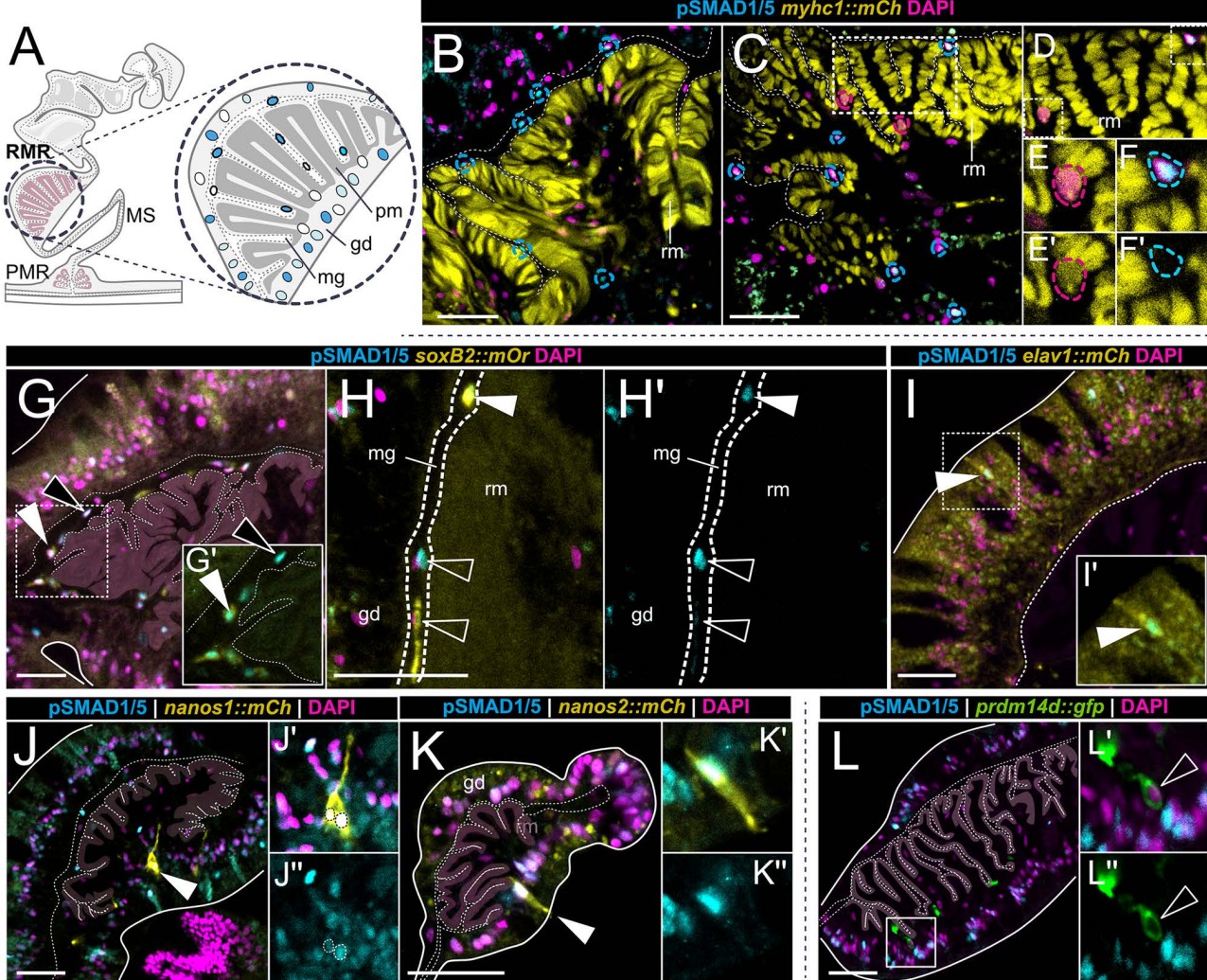

**Fig 3. BMP signaling is active in specific neuron types located in the retractor muscle region of the mesentery. (A)** Schematic overview of a mesentery cross-section, highlighting the retractor muscle region (RMR) within the neuro-muscular domain. **(B–K)** Immunostaining on cross-sections of the retractor muscle region for pSMAD1/5 (cyan) and nuclei (magenta) in transgenic reporter lines. **(B-F)** *myosin heavy chain 1* reporter expressing mCherry (yellow) (*myhc1::mCh*) showing pSMAD1/5-positive nuclei located in the mesoglea and gastrodermis, **(C-F)** detail, showing no overlap between pSMAD1/5-positive nuclei (cyan dashed outlines) and mCherry-positive nuclei of retractor muscle cells (magenta dashed outlines). **(G-H)** pSMAD1/5-positive nuclei overlap (white arrowhead) with mOrange in the *soxB(2)* reporter line (*soxB(2)::mOr*), **(I)** pSMAD1/5 activity overlaps with mCherry-positive cells in the *elav1* reporter line (white arrowhead). **(J)** *nanos1* reporter expressing mCherry (*nanos1:mCh*) shows pSMAD1/5 staining in *nanos1*-positive putative ganglionic cells (white arrowhead), **(K)** *nanos2* reporter expressing mCherry (*nanos2::mCh*) shows pSMAD1/5 staining in the *nanos2*-positive putative sensory cells (white arrowhead), **(L)** *prdm14d* reporter expressing GFP (*prdm14d::gfp*) does not show pSMAD1/5 staining in the *prdm14d*-positive cells (white outlined arrowhead). Immunostaining for pSMAD1/5 (cyan), DAPI (magenta), reporter protein (yellow or green). Scale bars 25 μm. White arrowheads indicate overlap, white outlined arrowheads indicate no overlap. gd—gastrodermis, pm—parietal muscle, mg—mesoglea, ep—epidermis.

*soxB2::mOr*-positive cells with differentiated sensory neuron or ganglionic neuron morphology, which were embedded in the epithelium of the parietal muscle region (Fig 4D–4D', white arrowheads). To further examine differentiated neuronal types, we analyzed the parietal muscle region in the reporter lines *elav1::mCh* and *nanos1::mCh* [28,43]. Both reporter lines label sensory and ganglionic neurons in both body layers, with *elav1::mCh* expression being especially pronounced

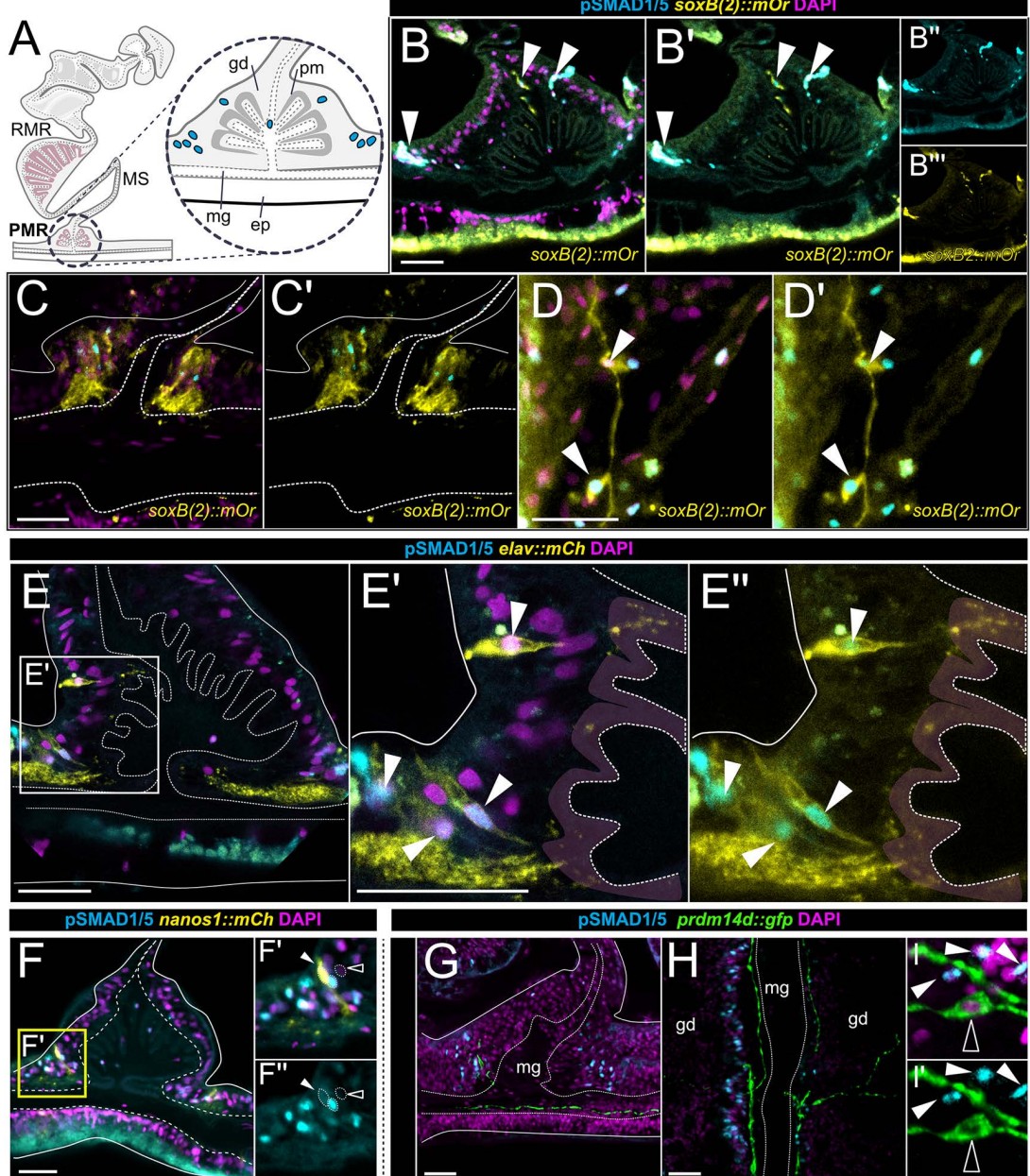

**Fig 4. BMP signaling is active in specific neuron types of the parietal muscle region of the mesentery. (A)** Schematic overview of a mesentery cross-section, highlighting the parietal muscle region (PMR) within the neuro-muscular domain. **(B–H)** Immunostaining for pSMAD1/5 (cyan) and nuclei (magenta) in the parietal muscle region of transgenic reporter lines in cross-sections or in lateral views. **(B-D)** mCherry-positive cells (yellow) in the *soxB(2)* reporter line (*soxB(2)::mCh*), located in the mesoglea and gastrodermis, partially overlap with pSMAD1/5 activity (white arrowheads), **(B-C)** in cross-sections and **(D)** lateral view. White arrowheads in **(E-F)** indicate pSMAD1/5-positive nuclei (cyan) in the epithelium of the parietal muscle region overlap **(E)** with mCherry (yellow) in the *elav1* reporter line (*elav1::mCh*) and **(F)** with mCherry (yellow) in the *nanos1* reporter line(*nanos1::mCh*). **(G–I')** No overlap (white outlined arrowhead) of pSMAD1/5 (cyan) and GFP (green) detectable in the *prdm14d* reporter line (*prmd14d::gfp*). gd – gastrodermis, pm – parietal muscle, mg – mesoglea, ep – epidermis.

in the neurite tracts [28,43]. In the *elav1::mCh* and *nanos1::mCh* reporter line, we discerned cells with sensory neuron-like appearance that were pSMAD1/5-positive (Fig 4E–4F'', white arrowheads). Notably, pSMAD1/5 was again present in cells not expressing fluorescent reporter proteins, suggesting the presence of other cell populations with active BMP signaling along the parietal muscle region (Fig 4E, 4F). Similar to the situation in the retractor muscle region, the *prdm14d::gfp* reporter line, which labels ganglionic neurons along the neurite tracts [40], showed no pSMAD1/5 in the *prdm14d::gfp*-positive cells in the parietal muscle region (Fig 4G–4I', white outlined arrowhead), similar to the situation in the retractor muscle region (Fig 3L). These findings suggest that BMP signaling is active in specific parts of the gastrodermal nervous system, where it is restricted to neuronal progenitors and distinct ganglion and sensory neurons of the mesentery, while it is absent from distinct subpopulations of the neuronal lineage in the same regions.

## BMP signaling is active in the medusozoan nervous system

Life cycles of medusozoan cnidarians contain a pelagic medusa stage, whose behavior is much more complex than that of a sessile polyp, indicative of a more elaborate nervous system. To test if we can also detect BMP signaling in the medusozoan nervous system, we performed pSMAD1/5 stainings on the cubozoan jellyfish *Tripedalia cystophora* at medusa stage (Fig 5), and on the early juvenile, the so-called metaephyra, of the scyphozoan jellyfish *Stomolophus sp.* (Fig 6). In *Tripedalia*, we performed co-stainings against pSMAD1/5 and α-Tubulin (αTub), the latter being known to label a large portion of the *Tripedalia* nerve net [45]. We found that α-Tubulin indeed marked large parts of the *Tripedalia* nerve net, including neurons in the umbrella (Fig 5B–5D') and the concentrated ganglia of the nerve ring running along the umbrella rim (Fig 5E–5F), but no αTub labeling was observed in the neurons located in the rhopalia, which are the Scyphozoa- and Cubozoa-specific sensory organs containing gravity-sensing statocysts and eyes (Fig 5I–5L). In the umbrella, BMP signaling was detectable in αTub-positive ganglionic neurons (Fig 5C) and cnidocytes (Fig 5D). Cnidocytes are a Cnidaria-specific, highly modified neuronal cell lineage of chemosensory stinging cells identifiable by their distinct crescent-shaped nuclei and the presence of a cnidocyst—the capsule containing the stinging thread (Fig 5D–5D'). In addition, BMP signaling was also pronounced in the so-called nematocyst batteries or nematocyst warts (S1 Fig), specialized cell clusters consisting of cnidocytes, supporting cells, sensory cells and neurons, located in the exumbrella [46]. pSMAD1/5 staining was especially pronounced in the cells of the nerve ring (Fig 5A and 5E–F'). Within the nerve ring, we find cells with pSMAD1/5-positive and pSMAD1/5-negative nuclei (Fig 5E–5F'), which suggests that, like in *Nematostella*, nuclear pSMAD1/5 marks only a subset of neurons in *Tripedalia*. The *Tripedalia* medusa possesses four rhopalia, each containing one statocyst with a heavy statolith (sta) and six eyes: a pair of pit eyes (pe), a pair of slit eyes (se), as well as a much more complex upper lens eye and a lower lens eye (ule, lle) (Fig 5G–5H'). Since the statolith is located off-center and the whole rhopalium is suspended on a flexible stalk, each rhopalium maintains its orientation in relation to the vector of gravity. Thus, each rhopalium has a clear top side and a clear bottom side. pSMAD1/5-positive cells were restricted to the top side of the rhopalium, including the rhopalium stalk (st), the cells at the stalk base and towards the upper lens eye (Fig 5J–5L). In the regions of the upper lens eye and pit eye, pSMAD1/5 activity is limited to cells closest to the stalk region. While neurons of the rhopalia are not marked by αTub, at least some of the pSMAD1/5-positive cells in the area between the stalk and the upper lens eye are likely to be neurons based on their location and cell morphology [47,48]. Some pSMAD1/5-positive cells in this area exhibit a high cytoplasm to nucleus ratio (Fig 5J and 5L white arrowheads), which allowed us to unequivocally identify them as the giant neurons previously described by others [47,48]. Moreover, in the pit eye and possibly also in the upper lens eye, a sector of the retina is composed of cells with pSMAD1/5-positive nuclei (Fig 5L). Thus, in the rhopalium, the location of the pSMAD1/5-positive cells is consistent with at least some of them having neural identities.

In the ephyra of *Stomolophus*, BMP signaling was broadly detectable in the central disc, the lappets and the rhopalia (Fig 6C, 6D). In the rhopalia, we observed BMP signaling in the basal (bs) and intermediate segment (in) but not the terminal segment (te) (Fig 6D). While the staining for different neuronal markers did not allow the identification of discrete

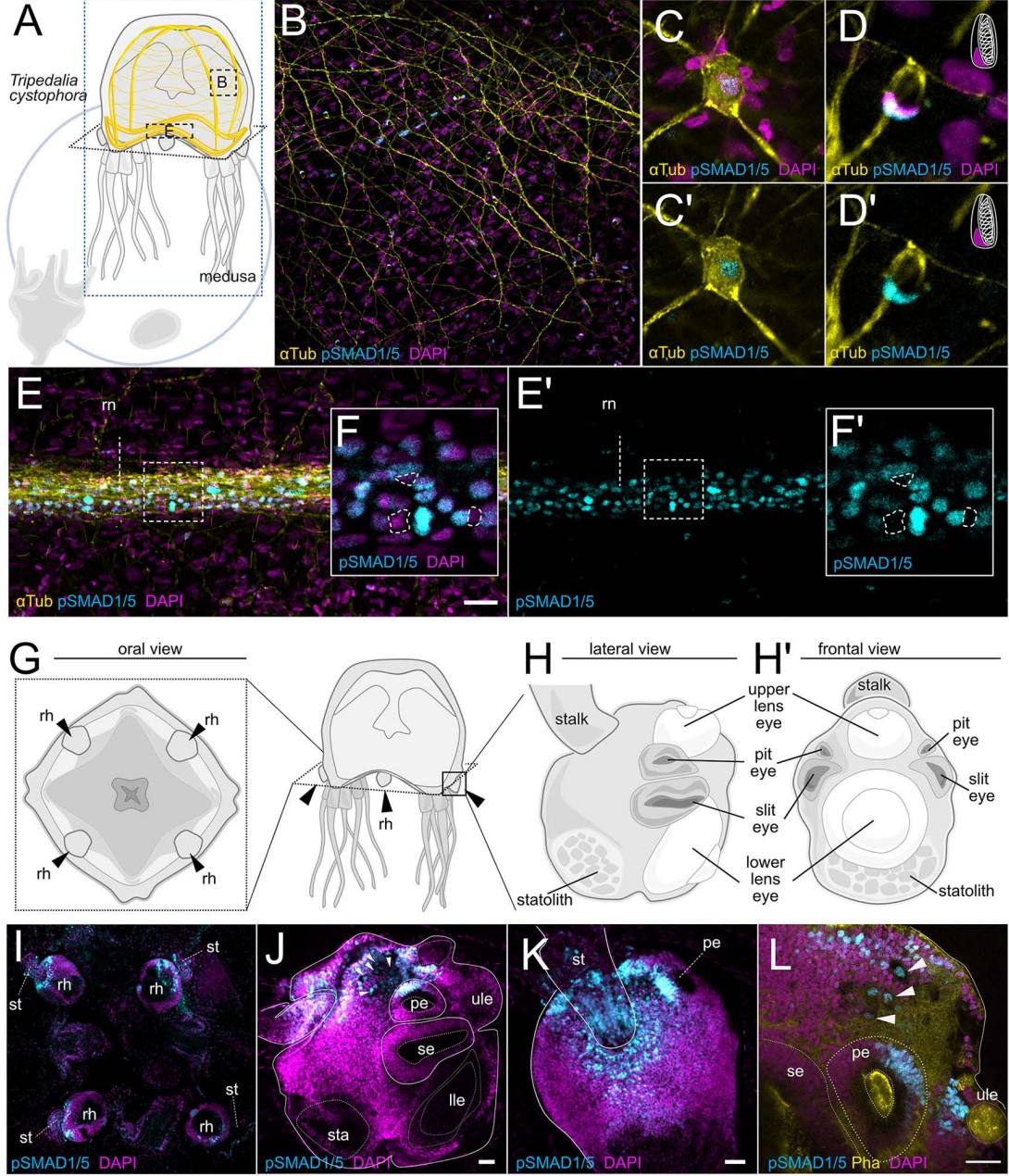

**Fig 5. In the box jellyfish *Tripedalia*, BMP signaling is active in ganglionic neurons of the umbrella, the ring nerve and in the rhopalium.**
**(A)** Life cycle schematic of *Tripedalia cystophora*. **(B)** α-Tubulin antibody staining highlighting the nerve net in the umbrella. **(C-C')** Detail of α-Tubulin/pSMAD1/5 double-positive neuron in the umbrella. **(D-D')** Detail of α-Tubulin/pSMAD1/5 double-positive cnidocyte in the umbrella. Note the crescent-shaped nucleus at the periphery of the unstained cnidocyst. **(E–F')** pronounced α-Tubulin and pSMAD1/5 staining of the ring nerve; **(F–F')** detail of pSMAD1/5-positive and -negative cells in the ring nerve. **(G)** Oral view of the umbrella, black arrowheads highlight the location of the rhopalia (rh). **(H)** lateral and **(H')** frontal detail of the rhopalium anatomy. **(I)** pSMAD1/5 activity in the rhopalia. **(J)** Lateral view of the rhopalium; pSMAD1/5 activity is restricted to the area between the stalk and the pit eye. **(K)** View from the top highlights pSMAD1/5 activity around the stalk. **(L)** Detailed view of the pit eye and the area around it; pSMAD1/5 is observed in a sector of the pit eye retina, the upper lens eye and some cells between the pit eye and the stalk. Arrowheads on **(J)** and **(L)** highlight the pSMAD1/5 activity in the giant neurons. Pha - phalloidin, rn - ring nerve, rh - rhopalium, st - stalk, pe - pit eye, se - slit eye, ule - upper lens eye, lle - lower lens eye, sta - statolith; Scale bar 25 μm.

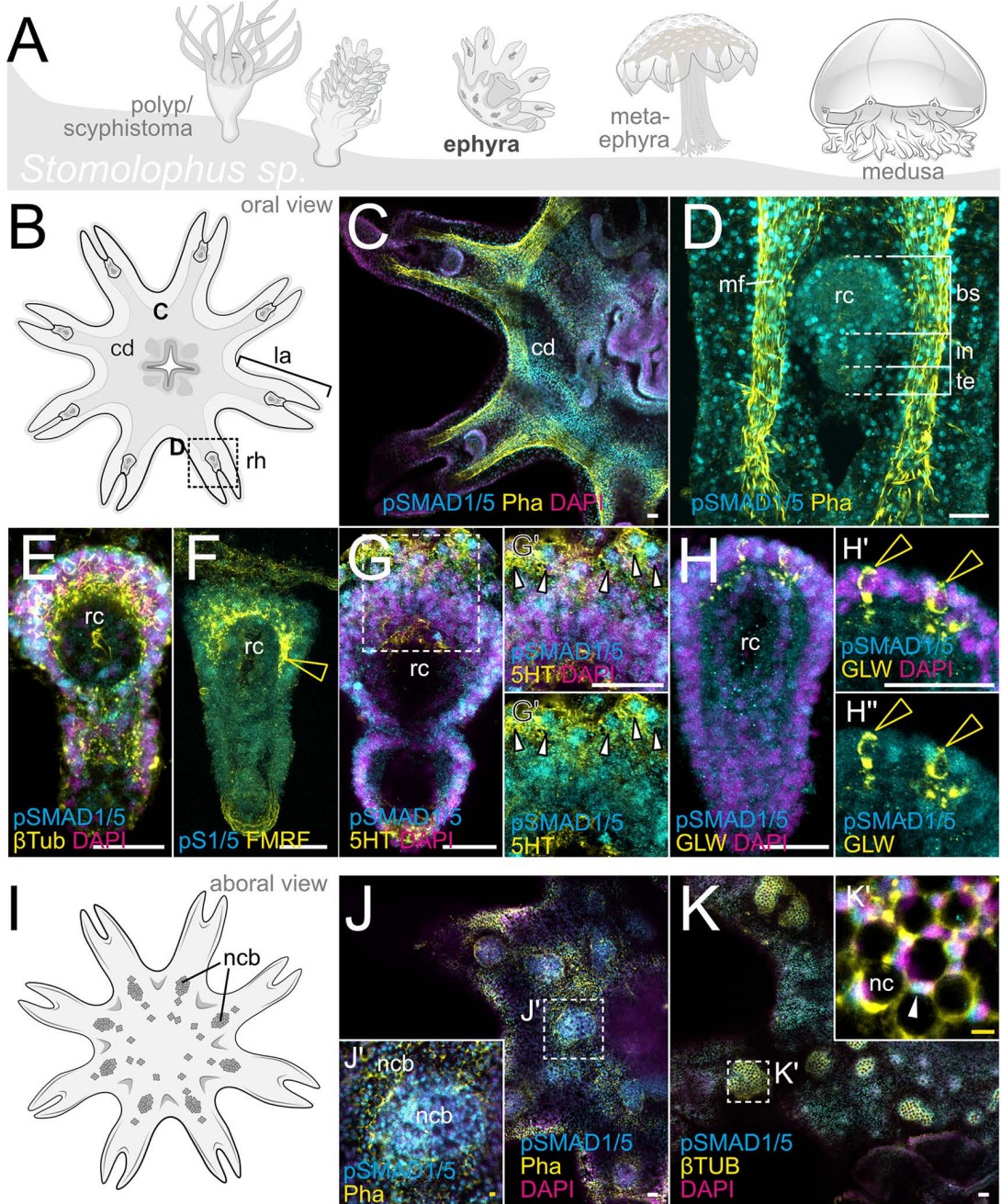

**Fig 6. BMP signaling activity in the ephyra of the cannonball jellyfish *Stomolophus sp.* is partially overlapping with neuronal markers labeling rhopalia and nematocyst batteries. (A)** Life cycle of *Stomolophus sp.* **(B)** Details of the ephyra (oral view) **(C)** BMP signaling (pSMAD1/5, cyan) is detectable in the central disc and rhopalia, phalloidin staining (Pha, yellow) highlights the muscle in the muscle field and lappets. **(D)** Detailed view of pSMAD1/5 activity in the lappet and rhopalium, phalloidin (Pha, yellow) mark muscle fibers of the lappet. **(E)** βTub broadly labels cells in the rhopalium, overlapping with both pSMAD1/5-positive and -negative cells in the rhopalar canal (rc) of the basal segment (bs). **(F)** FMRFamide (yellow) highlights the rhopalial canal of the basal segment. **(G-G")** Black outlined white arrowheads point at 5HT/pSMAD1/5-double-positive cells adjacent to the basal rhopalar segment. **(H-H")** GLWamide (yellow) labels distinct pSMAD1/5-negative cells in the basal rhopalar segment. **(I)** Ephyra aboral view, ncb - nematocyst batteries. **(J-J')** Pha (yellow) labeling f-actin shows reduced staining of pSMAD1/5-positive ncb. **(K-K')** β-Tub accumulates in the area of the ncb labeling nematocytes, ncb contain pSMAD1/5-positive nuclei and nematocyte capsules (black). rc -rhopalar canal, bs-basal segment, in-intermediate segment, te-terminal segment, ncb-nematocyst batteries, nc-nematocyst. Scale bar 25 μm (white) and 5 μm (yellow).

pSMAD1/5-positive, neuronal subpopulations in the rhopalium, anti-β-Tubulin broadly labeled rhopalial cells and cilia, with stronger staining in the pSMAD1/5-positive basal segment (Fig 6E). Similarly, FMRFamide signal was upregulated in the pSMAD1/5-positive area of the basal rhopalial segment (Fig 6F, hollow yellow arrowhead) and staining for serotonin (5-HT) broadly highlighted pSMAD1/5-positive cells of the lappet adjacent to the basal rhopalial segment (Fig 6G–6G", white arrowheads). In contrast, antibodies raised against GLWamide of the colonial hydrozoan polyp *Hydractinia* [49] labeled few, individual cells in the basal segment that were neighboring but not overlapping with pSMAD1/5-positive rhopalial cells (Fig 6H–6H", hollow yellow arrowheads). Similar to *Tripedalia*, *Stomolophus* forms nematocyst batteries (ncb), containing cnidocytes, supporting cells, sensory cells and neurons on the exumbrella surface at the aboral side of the ephyra. Areas of nematocyst batteries displayed reduced labeling by phalloidin while β-Tubulin staining was enriched compared to the surrounding epithelium (Fig 6J, 6K). In addition, many of the nematocyst battery cells were pSMAD1/5-positive (Fig 6J–6K'). While the exact identity of pSMAD1/5-positive nematocyst battery cells is not clear, multiple pSMAD1/5-positive nuclei again displayed a crescent-shaped morphology (Fig 6K, white arrowhead) suggesting that at least some of these cells are cnidocytes.

## BMP signaling attenuation in the adult *Nematostella* polyp results in the differential expression of neuronal regulators

To better understand the BMP signaling-dependent gene regulation in adult *Nematostella*, we suppressed BMP signaling in the mature polyp using the BMP receptor inhibitor K02288 (Fig 7A), dissected adult polyps into the head without tentacles (from female polyps), body wall with mesentery base (from female polyps), and isolated mesenteries (from female and male polyps separately), and performed bulk RNA-Seq in biological triplicates (Fig 7B). To enrich for direct rather than indirect downstream target genes, we wanted to keep inhibitor treatment as short as possible. We determined that 5 hours of K02288 treatment was the minimal duration sufficient to reliably inhibit BMP signaling in the polyp compared to the DMSO control as manifested by the disappearance of the nuclear pSMAD1/5 (Fig 7A). DeSeq2 analysis revealed effects of BMP inhibition in all tissues, with the differential expression (padj. of ≤ 0.05) of 203 genes in the head (145 down, 58 up), 233 genes in the body wall (206 down, 27 up), 56 genes in the female mesentery (53 down, 3 up), and 170 genes in the male mesentery (146 down, 24 up), comprising 489 unique genes differentially expressed across all tissues (Fig 7C). Among the different tissues, only 9 genes were differentially regulated in all tissue samples (Fig 7D and S1 Table). To check for shared BMP signaling target genes across developmental stages, we compared the K02288/DMSO treatment dataset of the adult polyp with two recently published datasets of BMP signaling targets during early development: the dataset of direct BMP signaling targets at Late gastrula and 4 day planula stage identified by pSMAD1/5 ChIP-Seq, as well as the bulk RNA-seq dataset of the 2d planula upon BMP2/4 morpholino knockdown (BMP2/4MO) [30]. Out of 210 pSMAD1/5 ChIP target genes at gastrula and 4d planula (see Materials and methods for details), 20 genes were also differentially expressed upon K02288 treatment. Seven of these 20 were transcription factors or co-factors (Fig 7E and S2 Table). 289 genes were differentially expressed upon both K02288 treatment and BMP2/4MO knockdown, and 12 genes were present in all three datasets (Fig 7E and S2 Table).

Both, in the RNA-seq datasets of BMP2/4 knockdowns and K02288 treatments, we identified various components of Wnt, FGF, TGFβ/BMP, Notch and MAPK signaling, of which some were also direct pSMAD1/5 target genes in *Nematostella* embryos (S2 Table). Previous works implicated multiple signaling pathways in nervous system development and maintenance in *Nematostella* [31], however, we cannot treat differential expression of these "broad-spectrum" regulatory genes as evidence for or against the involvement of BMP signaling in neural development. In contrast, some of the differentially expressed genes were clearly neural factors, suggesting neuronal regulation by BMP signaling across developmental stages. Previous characterization of the direct pSMAD1/5 target genes in Late gastrula and 4-day planula stage and of the genes differentially expressed upon morpholino knockdown of BMP2/4 yielded multiple putative or verified neuronal genes (S2 Table) [30]. Similarly, the dataset of K02288 treatments of adult polyps shows an overrepresentation

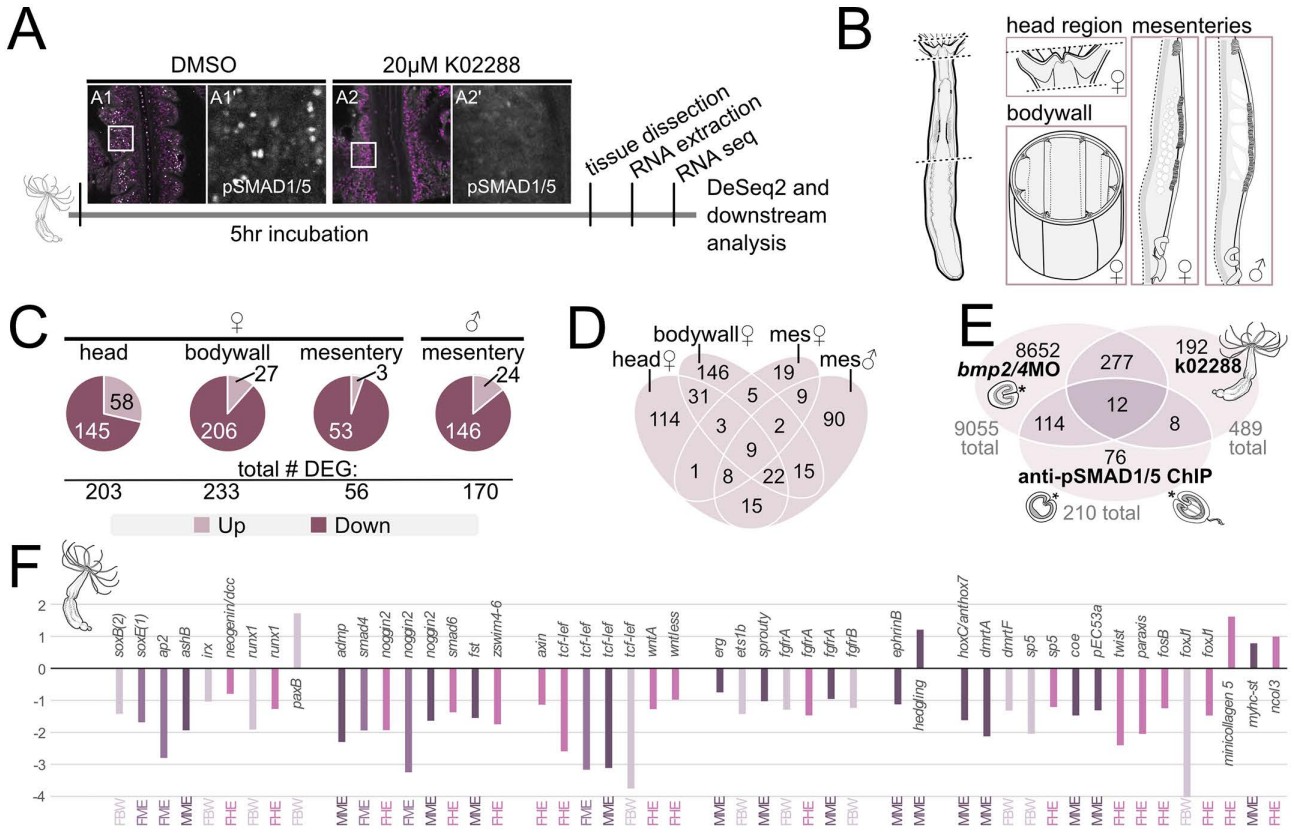

**Fig 7. Pharmacological suppression of BMP signaling in the adult *Nematostella* polyp reduces the expression of developmental and neuronal regulators. (A)** Schematic of BMP signaling inhibition in the adult polyp using K02288 treatment and downstream processing and analysis. **(B)** Schematic of tissue dissection for tissue-specific bulk RNA-seq **(C)** pie charts illustrating the number of down- and upregulated genes upon K02288 treatment in individual tissues. **(D)** Venn diagram showing the relationships of differentially expressed genes (DEGs) between sets of tissues. **(E)** Venn diagram illustrating the relationships between the combined set of DEGs in K02288 treated adults and previously published data sets of DEGs in the 2-day planula upon *bmp2/4* morpholino knockdown and direct pSMAD1/5 targets at Late gastrula and 4d planula [30]. **(F)** Bar plot showing log2 fold change (padj. < 0.05) of selected differentially expressed genes in adult polyp tissues (FBW-female body wall, FME-female mesentery, MME-male mesentery, FHE-female head region) upon K02288 treatment. Numerical data for the bar plot can be found in the sel.K02288 sheet of the S1 Table.

of the GO-terms for neurogenic processes among the differentially expressed genes: 332 of the 489 unique GO-terms are neurogenesis-related (S1 Table and S2 Fig). The K02288 dataset of differentially expressed genes contains various putative or verified neuronal markers (*soxB(2), soxE1, ashB, paxB, sp5, fosB, ap2, runx, irx, dmbx, foxJ1, zswim4-6, ephrinB, c-ski,* etc.; S1 Table and Fig 7F), the majority of which are downregulated upon attenuation of the BMP signaling.

## Neural lineage markers are downregulated by BMP signaling attenuation in the *Nematostella* larvae

Previous analyses suggested that BMP signaling has both positive and negative effects on neurogenesis in the planula larvae [32]. We searched for differentially expressed neural regulators in the BMP2/4MO RNA-seq data set, revealing multiple neuronal genes, most of which were significantly downregulated (S2 Table). We analyzed the expression patterns of selected neuronal genes upon the suppression of BMP signaling by BMP2/4 morpholino or K02288 treatment using in situ hybridization. First, we analyzed the expression of *soxC* and *soxB(2),* marking neuronal progenitors [42,43], upon BMP2/4MO knockdown, resulting in the complete loss of the pSMAD1/5 gradient, and gdf5-lMO knockdown, reducing graded pSMAD1/5 activity [33]. The expression of both *soxC* and *soxB(2)* was strongly reduced in bmp2/4MO (Fig

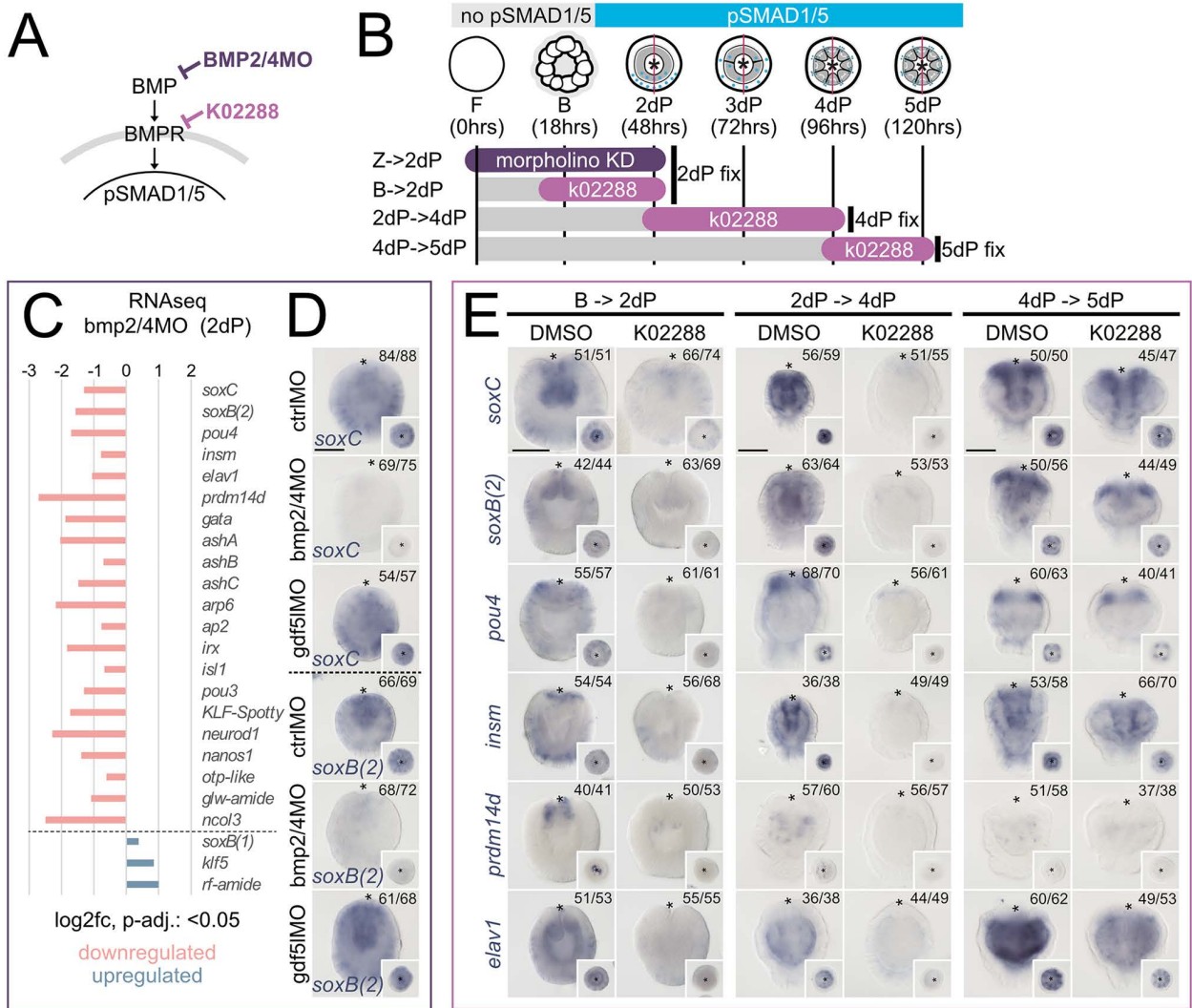

**Fig 8. Neuronal markers are downregulated upon BMP signaling inhibition in *Nematostella* planula. (A)** Schematic of the BMP signaling pathway and how it is affected by BMP2/4 morpholino oligo and pharmacological inhibition of the BMP receptor with K02288. **(B)** Timeline of *Nematostella* early development and BMP signaling dynamics from fertilization **(F)**, Blastula **(B)** and 2-5 day Planula larvae (2dpP-5dP). The gray/cyan-colored bar indicates the absence/presence of pSMAD1/5 activity. Duration and fixation time points of morpholino knockdown experiments (purple) and K02288 treatments (lilac). **(C)** Bar plot of expression changes (log2 fold, padj. < 0.05;) of selected neuronal marker genes in 2 day planula upon bmp2/4MO knockdown by RNA-seq analysis [30], see Materials and methods. **(D)** Expression analysis of *soxC* and *soxB(2)* expression upon ctrlMO, bmp2/4MO, and gdf5-IMO knockdown at 2 day planula stage by in situ hybridization. **(E)** Selected neuronal marker expression upon K02288 treatments at 2 day planula, 4 day planula and 5 day planula stage visualized by in situ hybridization. Asterisks indicate blastopore, scale bars 100μm. Numerical data for the bar plot can be found in the sel.BMPMO_neuro sheet of the S2 Table.

8C, 8D). In contrast, *soxC* and *soxB(2)* were not affected by gdf5-IMO (Fig 8D), suggesting that either their expression requires input specifically from BMP2/4-mediated and not GDF5-like-mediated signaling or that reduced BMP signaling is sufficient to maintain their normal expression. Then, we analyzed the effect of the chemical inhibition of BMP signaling using the K02288 inhibitor, spanning different time windows: i) from Blastula (16–18 hpf) to 2d planula stage (**B->2dP**), which overlaps with the time of the pSMAD1/5 gradient formation; ii) from 2d to 4d planula stage (**2dP->4dP**), when the pSMAD1/5 activity gradually switches from graded to dispersed and becomes located in the mesenteries, corresponding

to the time of gastrodermal patterning and mesentery formation; and iii) from 4d to 5d (**4dP->5dP**), after the directive axis has been fully patterned and pSMAD1/5 is active in the mesenteries. For the different time windows of treatment, we again examined the expression of differentially expressed neural markers from the BMP2/4MO dataset, including *soxC* and *soxB(2),* as well as *insulinoma (insm),* which is expressed in neurosecretory and neuroglandular populations [50], *prdm14d* as marker for neural progenitors and ganglion neurons in the gastrodermis [40] and *pou4* and *elav1* as neuronal differentiation markers of gastrodermal and epidermal subpopulations [28,51]. K02288 treatments from blastula to 2d planula resulted in strong downregulation of all analyzed marker genes (Fig 8E), concurring with the results of the BMP2/4MO in situ and RNA-seq data. Similarly, BMP inhibition starting at 2d planula until 4d planula stage, strongly reduced neuronal marker genes (Fig 8E). In comparison, K02288 treatments from the 4d to 5d planula, which is after the directive axis has been patterned, and mesenteries have been formed, displayed only a mild reduction of neural marker expression (Fig 8E), suggesting that the neuronal regulation becomes less sensitive to BMP signaling input in the fully patterned late planula. While early studies indicated a combination of positive and negative effects of BMP signaling on neural patterning in the planula, our data support a predominantly pro-neural influence of BMP signaling in the nervous system of the *Nematostella* larva.

### Putative neural lineage markers are downregulated by BMP signaling attenuation in the metaephyra of the scyphozoan *Stomolophus*

Our immunohistochemical analyses suggested that BMP signaling is active in the neurons and cnidocytes of the cubozoan and scyphozoan jellyfish. Therefore, we asked what effect BMP signaling had on the expression of the putative neuronal and cnidocyte markers in the scyphozoan jellyfish *Stomolophus*. We performed RNA-Seq on 4 biological replicates per treatment of K02288- and DMSO-treated *Stomolophus* metaephyrae, assembled *Stomolophus sp*. transcriptome de novo using TRINITY, identified *Stomolophus* transcripts differentially expressed upon K02288 treatment with DeSeq2, and extracted their potential *Nematostella* orthologs with OrthoFinder. Using the published single-cell RNA-Seq dataset [37], we then identified which of the *Nematostella* orthologs of the differentially expressed *Stomolophus* genes were specifically expressed in neuronal or cnidocyte lineages. Due to limited material, we were unable to validate the expression of the *Stomolophus* orthologs of these genes by in situ hybridization and scRNA-Seq in metaephyrae, but we considered them to be putative neuronal or cnidocyte genes in *Stomolophus* and compared changes in their expression upon BMP signaling suppression to the changes in the expression of their *Nematostella* orthologs upon similar treatments (Fig 9A, 9B).

DeSeq2 analysis revealed 7,661 *Stomolophus* transcripts differentially expressed in the K02288-treated and DMSO-treated metaephyra (padj <0.05). 5,803 of these 7,661 DEGs had a *Nematostella* ortholog according to OrthoFinder, corresponding to 4,862 *Nematostella* gene models (S3 Table). Multiple *Stomolophus* transcript hits to a single *Nematostella* gene model were likely due to truncated transcripts after de novo assembly matching different parts of the same *Nematostella* gene model, uncollapsed isoforms, or *Stomolophus*-specific gene duplications. We then asked whether any of the *Nematostella* orthologs of the differentially expressed *Stomolophus* transcripts were differentially expressed in the K02288 treated adult *Nematostella* polyp tissues and found 144 such DEGs (Fig 9C and S3 Table). We plotted their expression across *Nematostella* cell states using the published single-cell RNA-Seq dataset [37] and, along with many genes with broad expression, found 4 genes expressed specifically in neuronal cells and 11 specifically expressed in the cnidocytes (Fig 9C and S1 Plot). All 4 neuronal-specific genes were downregulated upon K02288 treatment in *Nematostella* and in *Stomolophus*, in line with the pro-neural function of BMP signaling in both species (Fig 9C and S3 Table). In contrast, 10/11 cnidocyte markers differentially expressed in both species were downregulated in *Stomolophus* metaephyrae but upregulated in the *Nematostella* polyp (Fig 9C and S3 Table), which is also in line with *Stomolophus* cnidocytes being pSMAD1/5-positive and *Nematostella* cnidocytes being pSMAD1/5-negative (Fig 6K', [36]).

Since metaephyra is a developing juvenile rather than a steady-state animal, we also intersected *Nematostella* orthologs of the *Stomolophus* differentially expressed transcripts with the set of DEGs from 2d old *Nematostella* larvae

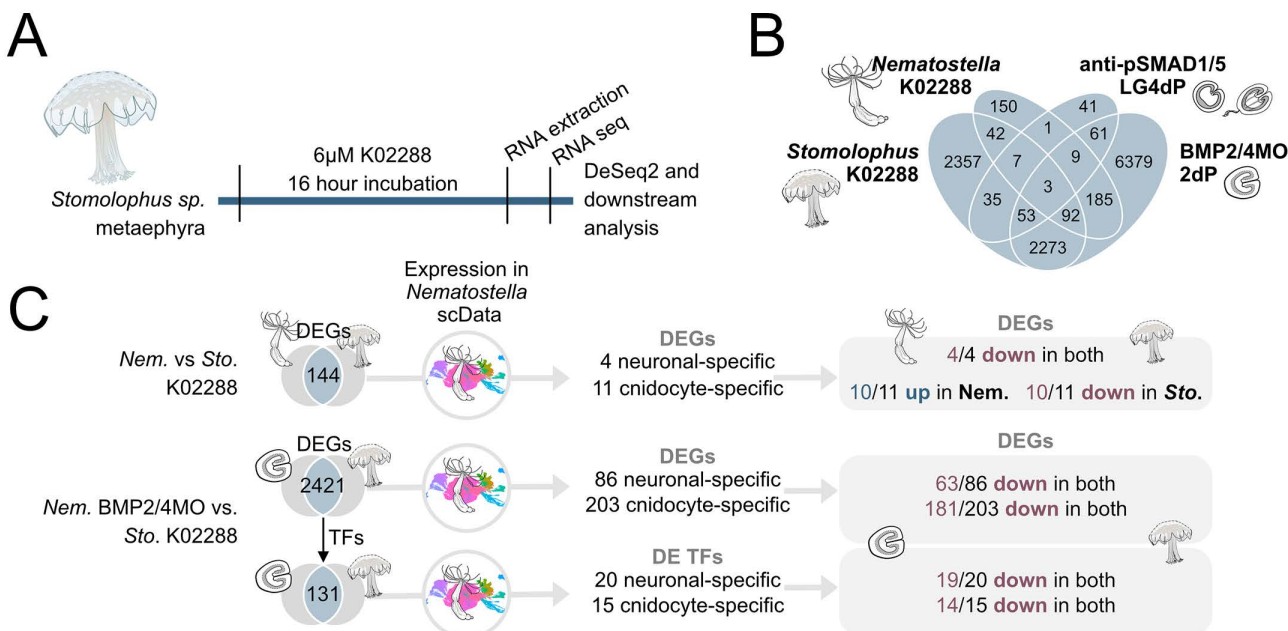

**Fig 9. Pharmacological suppression of BMP signaling in the *Stomolophus* metaephyra reduces the expression of putative neuronal and cnidocyte-specific genes. (A)** Schematic of BMP signaling inhibition in metaephyra using K02288 treatment and downstream processing and analysis. **(B)** Venn diagram illustrating the relationships between the combined set of DEGs in K02288 treated *Stomolophus* metaephyra, K02288 treated *Nematostella* polyps and previously published *Nematostella* datasets of DEG genes in the 2-day planula upon *bmp2/4* morpholino knockdown and direct pSMAD1/5 targets at Late gastrula and 4d planula [30]. **(C)** Intersecting DEGs of K02288-treated *Nematostella* and *Stomolophus*, as well as of *Nematostella* BMP2/4 2 day morphants and K02288-treated *Stomolophus*. *Nematostella* DEGs and DE TFs were plotted using *Nematostella* developmental single-cell RNA-Seq data [37] to identify putative neuronal- and cnidocyte-specific gene orthologs in *Stomolophus*.

with BMP signaling abolished by BMP2/4 morpholino knockdown [30]. Here, 2421 genes were found to be differentially expressed in both species (Fig 9C and S3 Table). This very extensive DEG overlap is likely due the presence of not only direct but also indirect BMP target genes in the shared DEG list caused by the prolonged BMP signaling suppression in the *Stomolophus* metaephyra and morphant *Nematostella*. We analyzed the expression of these 2421 genes across the different *Nematostella* cell states and identified 86 neuronal-specific and 203 cnidocyte-specific genes (Fig 9C and S2 and S3 Plots). Upon BMP signaling inhibition, 63 out of 86 neuronal genes and 181 out of 203 cnidocyte genes were downregulated in both species (Fig 9C and S3 Table). This strongly suggested that during development, both the neural and the cnidocyte formation is under positive BMP control in both species. Next, we selected all 131 transcription factors (TFs) present among the 2,421 DEGs and identified 20 neuronal-specific TFs and 15 cnidocyte-specific TFs (Fig 9C and S4 Plot). 19/20 neuronal-specific and 14/15 cnidocyte-specific TFs were downregulated upon BMP signaling suppression in both species (Fig 9C and S3 Table). Thus, at the level of TFs, positive regulation of neuronal and cnidocyte differentiation by BMP signaling in *Nematostella* and in *Stomolophus* became even more pronounced.

Taken together, we see a clear positive regulation of BMP signaling of neuronal genes in *Nematostella* and their putatively neuronal *Stomolophus* orthologs. In *Nematostella*, we also find a striking change in the modality of the response of the cnidocyte-specific genes to BMP signaling: from positive in the larva to negative in the adult, which suggests that BMP signaling—at least initially—positively regulates the development of the whole cnidocyte and neuronal lineage, which have been shown to share a common origin [43].

## BMP signaling is active in the neurogenic domain of a spiralian

Arrow worms (Chaetognatha) are planktonic predators named after their dart-shaped body and impressive jaw apparatus. Phylogenomic analyses place them within or as a sister group to Gnathifera, within the superphylum Spiralia [52,53]. Chaetognaths possess a brain composed of several ganglia and a centralized ventral nervous system (ventral nerve center). During embryogenesis, the ventral CNS develops from paired lateral neuroectodermal cell clusters, where cells proliferate to form "lateral somata clusters" and eventually fuse below the gut [54]. In line with observations in other protostomes, we find dorsal expression of *bmp2/4* and ventral expression of *chordin* in the midgastrula of *Spadella cephaloptera* (S3 Fig). Antibody staining shows a dorsal-to-ventral gradient of nuclear pSMAD1/5 in the gastrula and early elongation stages (Fig 10A–10C'). Importantly, pSMAD1/5 signal is observed in the neuroectodermal regions that later contribute to lateral somata clusters, which will eventually fuse ventrally to form a CNS (Fig 10B–10C'). In the hatchling and juvenile, the pSMAD1/5 staining shows broader distribution, including in mesodermal and epidermal derivatives. Nonetheless, BMP signaling remains prominent in the lateral somata, as well as in a number of dorsal sensory organs such as the *pax6*-expressing corona ciliata in the head and ciliary sensory organs of the trunk (Fig 10D–10E', [54]). These results show that parts of the central nervous system of *Spadella*, particularly the dorsal regions of the neuroectoderm, are located within the pSMAD1/5-positive domain, and that BMP signaling remains active in the ventral nerve center and in the forming sensory organs later in development.

## Discussion

### BMP signaling promotes neural gene expression in *Nematostella*

Regulatory functions of BMP signaling are manifold in Bilateria, and this multifunctionality also extends to their closest relatives, the cnidarians [36]. In this study, we analyzed the activity of BMP signaling in a part of the gastrodermal nervous system of the sea anemone *Nematostella.* The previously described overall neural architecture of the polyp includes multiple types of sensory and ganglion neurons with stereotypical placement in the body column [26–28,31]. Different neuron types are present in the two body layers, the epidermis and the gastrodermis, and can be generated by the respective tissue layer [28]. In the gastrodermis of the polyp, subsets of neurons are regionalized and accumulate in proximity to the longitudinal musculature in the mesentery, which is especially pronounced in the longitudinal neurite tracts along the parietal muscle [26,28]. In this region of neural condensations, we observed pronounced BMP signaling activity, partially overlapping with specific subsets of neurons and neuronal progenitors (Fig 11A). We found elevated activity of BMP signaling in neurons of the gastrodermis, specifically within the retractor and the parietal muscle regions. BMP signaling was overlapping with neurons expressing *elav1* and *nanos1*, but it was absent from other neurons located in the same regions, e.g., expressing *prdm14d*. BMP signaling was also present in two types of *soxb(2)*-positive neurons, firstly, in neurons with differentiated morphology in the epithelium, and secondly, in basiepithelial cells, likely representing neuronal progenitor cells, based on their localization in the mesoglea. In line with that, analysis of the single-cell RNA-Seq dataset shows the expression of BMP signaling components in the pSC.NPC.1 subset of putative stem cells/neuronal progenitor cells (S4 Fig). Notably, the occurrence of pSMAD1/5-positive basiepithelial cells, which were also *soxb(2)*-positive in most cases and rarely *soxb(2)*-negative, seemed to be restricted to the neuro-muscular mesentery region, while basiepithelial cells in other parts of the body were always pSMAD1/5-negative. These observations are especially interesting in light of the recent findings showing that basiepithelial *vasa2/piwi1* double-positive cells represent a population of putative multipotent stem cells of the mesentery, which contributes to at least a portion of the neuroglandular/-secretory repertoire of the gastrodermis in the juvenile and adult polyp [41]. The progeny of these *vasa2/piwi1*-positive cells include *soxB(2)*-positive and *prdm14d*-positive neuronal progenitors, as well as *elav1*-positive neural and *insm1*-positive neuroglandular cells [41]. Previously, we could determine that basiepithelial vasa2-positive cells located in the medial mesentery and septal filament have no detectable BMP signaling activity, indicating BMP signaling is absent from putative multipotent stem cells [36].

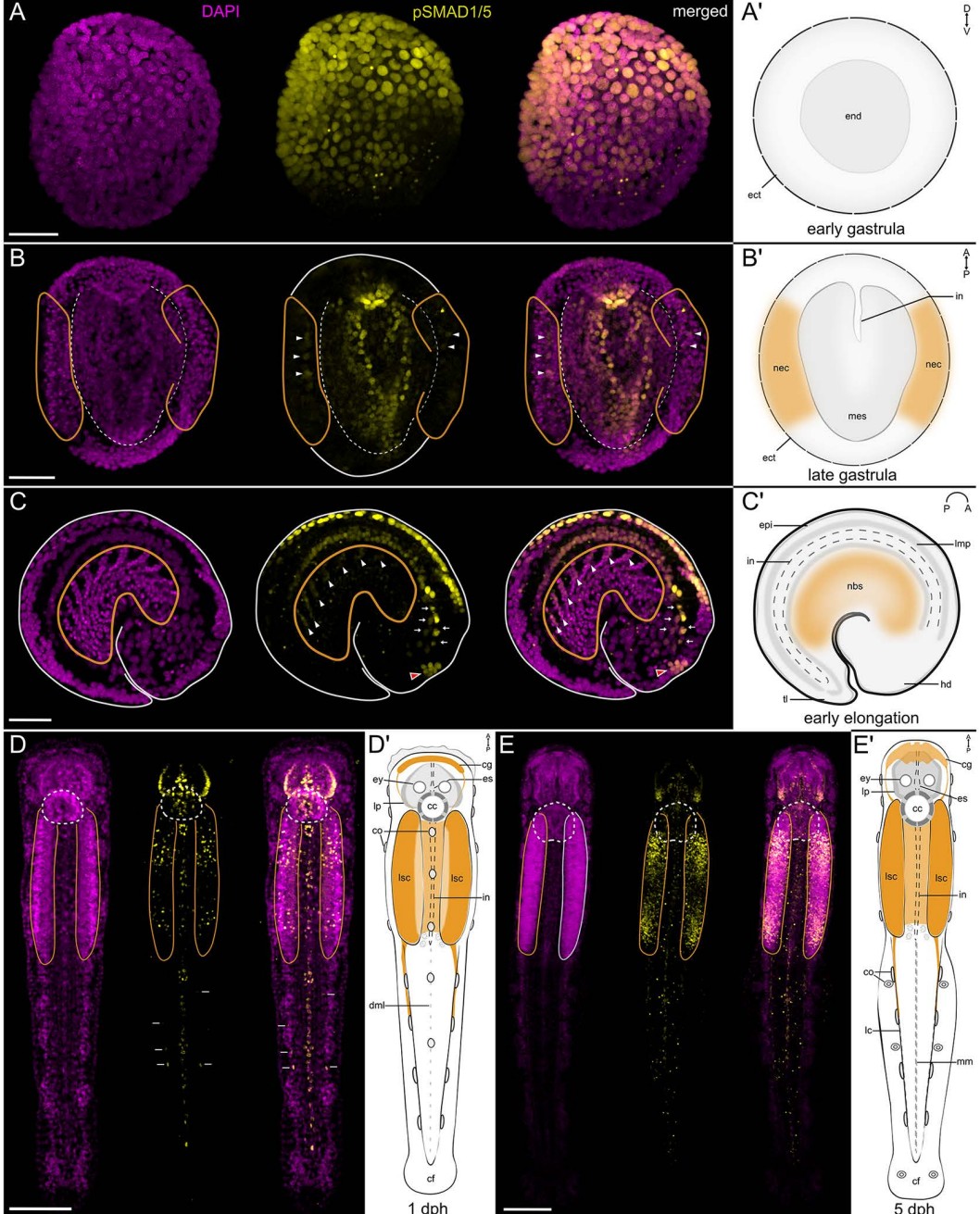

**Fig 10. BMP signaling is active in the developing nervous system of the chaetognathe *Spadella*. (A-A')** Transverse view of an early gastrula **(A)**, and schematic representation of its anatomy **(A')**. Nuclear pSMAD1/5 forms a dorsal-to-ventral gradient. pSMAD1/5-positive nuclei are observed in the dorsal ectoderm and endoderm (ect, end on **A'**). **(B-B')** Dorsal view of a late gastrula **(B)**, and schematic representation of its anatomy **(B')**. Nuclear pSMAD1/5 is detected in the dorsal mesoderm (mes on **B'**) and in some neuroectodermal nuclei (arrowheads; nec on **B'**). **(C-C')** Lateral view of the early elongation stage embryo **(C)** and schematic representation of its anatomy **(C')**. pSMAD1/5-positive nuclei are present in the dorsal epidermis (epi on **C'**), the developing digestive tract (in on **C'**), including the presumptive foregut (arrows), dorsal neuroblasts (white arrowheads) of the developing ventral nerve center (nbs on **C'**), and presumptive progenitor cells of the inner corona ciliata (red arrowheads). **(D, D')** Dorsal view of the *Spadella* hatchling **(D)** and schematic representation of its anatomy **(D')**. **(D)** In the head of the hatchling, pSMAD1/5-positive nuclei are detected in the esophagus (es on **D'**), lateral plate (lp on **D'**). Additional signal is observed in sensory structures: corona ciliata (inner cells; cc on **D'**) and dorsal ciliary tuft/fence organs (co on **D'**). pSMAD1/5-positive nuclei are also present in the dorsal epidermal midline extending along the trunk and tail (dml on **D'**), the intestine (in on **D'**),

 

and lateral somata clusters of the ventral nerve center (lsc on **D'**). Some pSMAD1/5-positive nuclei of unknown identity are also located lateral to the tail midline (arrows). **(E, E')** Dorsal view of the juvenile **(E)** and schematic representation of its anatomy **(E')**. pSMAD1/5-positive nuclei remain present in the esophagus, lateral plate, and intestine (es, lp, in on **E'**), and are particularly prominent in the lateral somata clusters of the ventral nerve center (lsc on **E'**). Expression is also detected in mesoderm-derived lateral cells (lc on **E'**) and mesenterial cells (mm on **E'**). pSMAD1/5 signal is shown in yellow, nuclei are stained with DAPI (magenta). Orange outlines demarcate the neuroectoderm **(B)** and the developing ventral nerve center **(C–E)**. The dashed outline in **(B)** marks the developing mesoderm and intestine. Dashed circles in **(D)** and **(E)** indicate the corona ciliata. Scale bars, 50 µm **(A–C)** and 100 µm **(D, E)**. Abbreviations: cc, corona ciliata; cf, caudal fin; cg, cerebral ganglion; co, ciliary tuft/fence organs; dml, dorsal epidermal midline cells; ect, ectoderm; end, endoderm; epi, epidermis; es, esophagus; ey, eye; hd, head; in, intestine; lc, lateral cells; lmp, longitudinal muscle cell precursors; lp, lateral plate cells; lsc, lateral somata cluster; mes, mesoderm; mm, mesenterial cells; nbs, neuroblast of the developing VNC; nec, neuroectoderm; tl, tail bud.

In the context of the proposed model for cell lineage relationships in the gastrodermis [41], we can speculate that BMP signaling activity may correspond to specific branches of the neuroglandular lineage (Fig 11B). BMP signaling may be involved in the decision-making of neuronal progenitors and their derivatives, which is supported by the transcriptional response upon BMP signaling inhibition.

Our bulk RNA-seq analysis of K02288 treated adult tissue revealed differential expression of multiple neuronal transcription factors including *soxB(2)*, *irx*, *ap2*, *ashB* and *paxB* [55–58]. Several gene targets in the transcriptomic data from the adult also overlapped with target genes of our previously published RNA-seq data and anti-pSMAD1/5 ChIP data from early stages of development [30]. In the datasets from the early developmental stages, we found that many neuronal genes were differentially regulated in the 2d planula upon BMP signaling inhibition by BMP2/4 morpholino-mediated knockdown (BMP2/4MO) and/or direct gene targets of BMP signaling at Late gastrula and 4d planula stage. Based on the overlap of these datasets, BMP signaling appears to regulate neuronal genes both directly and indirectly across different stages of *Nematostella* development (Fig 11C). These findings are in agreement with previous analyses of the embryo and the planula larvae suggesting that BMP signaling is involved in patterning and maintenance of the planula nervous system [32]. Using BMP2/4 morpholino knockdown and reverse transcription (RT)-PCR, Watanabe et al.described the downregulation of genes encoding for a group of achaete-scute and atonal-related basic helix-loop-helix (bHLH) proteins and of the neuropeptides *rf-amide* and *glw-amide* [32]. Our RNA-seq analysis of BMP2/4MO planulae yielded similar results, expanding our list of neurogenic factors, the majority of which proved to be downregulated in the absence of BMP signaling. Downregulated genes include *soxb(2)*, marking neuronal progenitors that give rise to nematocytes, sensory and ganglion neurons [42,57], as well as *soxC*, an upstream regulator of *soxB(2)* [43], which we have previously identified as a direct target gene of pSMAD1/5 [30]. Moreover, among the downregulated genes we find markers of neural progenitors and neural differentiation (*pou4, insm, elav1, prdm14d, nanos1, otp;* [28,50,51,59,40]), bHLH family members *(ashA, ashB, ashC, ashD, arp6, ato7;* [32,55,57]) and transcription factors associated with neurogenesis in bilaterians (*gata, soxE1, ap2, irx, isl1, pou3, runx;* [31,60]. In accordance, RNA detection by in situ hybridization in 2d planula upon BMP2/4 morpholino knockdown or K02288 BMP receptor inhibition showed significant downregulation of neurogenic factors. Notably, low levels of marker gene expression were still detectable, indicating that neurogenesis was not completely abolished, supporting the idea that BMP signaling is not required for the initiation of neurogenesis in the early embryo. Taken together, our data suggest a predominantly pro-neural role of BMP signaling in *Nematostella* larva and adult.

## BMP signaling activity overlaps with neuronal populations in Medusozoa

To determine if BMP signaling also plays a role in the diffuse nervous system of medusozoans, we analyzed the activity of BMP in the box jellyfish *Tripedalia* and the cannonball jellyfish *Stomolophus*. Co-stainings of pSMAD1/5 and α-tubulin in *Tripedalia* revealed BMP signaling activity in different parts of the diffuse nerve net of the medusa, highlighting ganglion neurons in the umbrella and the ring nerve. Similar to the situation in *Nematostella*, where we observed that pSMAD1/5 activity differs between *elav*1-positive and *prdm14d*-positive neurons, we found both pSMAD1/5-positive and -negative

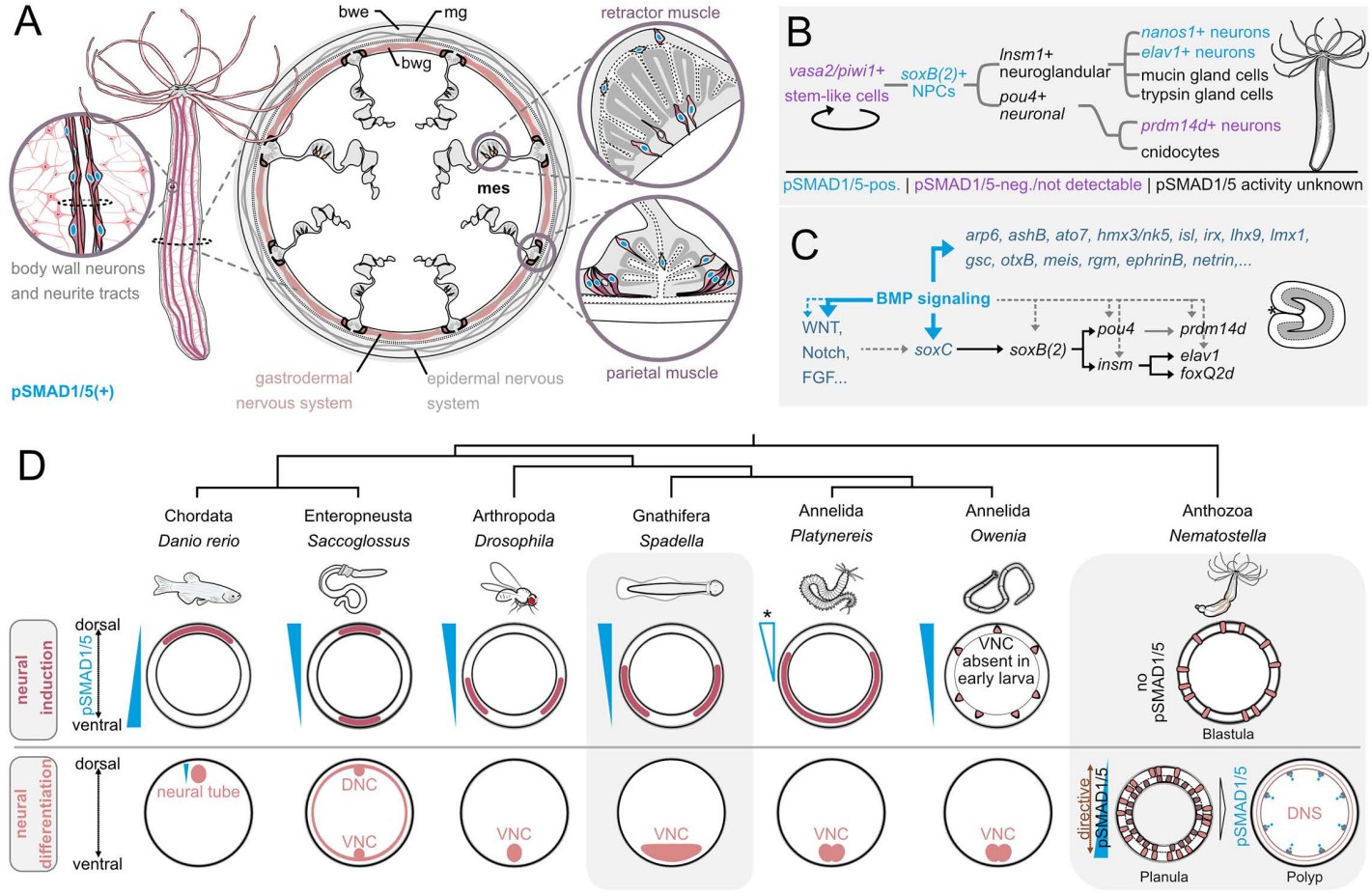

**Fig 11. Summary of BMP signaling in the neuronal lineage of *Nematostella* and comparison of neuronal development and graded BMP signaling activity in different Bilateria and *Nematostella*.** (A) Localization of pSMAD1/5-positive and -negative neurons in the gastrodermal nervous system. (B) A model for BMP activity during neuronal differentiation in adult polyps. (C) BMP signaling-mediated regulation of neuronal genes in the planula larvae. (D) Comparative overview of phylogenetic relationships, regions of neural induction and nervous system anatomy in Bilateria and *Nematostella*. bwe-body wall epidermis, mg-mesoglea, bwg-body wall gastrodermis, mes-mesentery, DNC-dorsal nerve cord, VNC-ventral nerve cord, DNS-diffuse nervous system.

neurons in the ring nerve of *Tripedalia,* indicating the presence of distinct neuronal subsets. In the ring nerve, the presence of specific neuronal populations labeled by different antibody combinations was demonstrated in the hydromedusa *Cladonema* [61] and in *Tripedalia* [45]. In *Tripedalia* and *Stomolophus*, we detected BMP signaling activity in distinct parts of the light- and gravity-sensing rhopalia. Similarly, BMP signaling was also active in the rhopalia of another scyphozoan, the moon jellyfish *Aurelia* (S5 Fig). The rhopalial nerve net features distinct subsets of regionalized sensory and ganglion cells [45,47,48,62,63]. In *Stomolophus* and *Aurelia*, BMP signaling activity was more pronounced around the rhopalial canal compared to the distal segment (S5 Fig). The regionalization of BMP signaling was even more pronounced in the rhopalia of *Tripedalia*, where areas of pSMAD1/5 activity were restricted to the region adjacent to the rhopalium stalk, pit eyes and the upper lens eye. Outside the eyes, pSMAD1/5 activity is found in the areas known to be populated by specific neurons, including PCNA-positive neurons innervating the pit eye and the upper lens eye, the giant neurons, FMRF-positive neurons around the stalk, but not in the flank neurons [48,47]. Transcriptomic analysis of the putative neuronal genes in *Stomolophus* showed that a vast majority of them, including *soxC*, *anr7*, *ashA*, *ashC*, *ashD*, and *neurogenin* are

positively regulated by BMP signaling. Curiously, in all three medusozoan species, *Tripedalia*, *Stomolophus* and *Aurelia*, we identified pSMAD1/5 activity in the stinging cells (cnidocytes, S5 Fig), a highly specialized cell type of the neurosecretory lineage across Cnidaria [43,60]. In line with that, multiple genes responsible for cnidocyte differentiation in *Nematostella* (*paxA, znf845, nr12, pou4, prdm6, prdm13, jun1, fos1* and minicollagens *ncol1, ncol3, ncol5;*) were suppressed upon BMP signaling inhibition in *Stomolophus*. In contrast, in the anthozoan *Nematostella,* where mature cnidocytes are always pSMAD1/5-negative [36], our transcriptome analyses show that the role of BMP signaling switches from pro-cnidogenic in the larvae to anti-cnidogenic in the adult. In summary, both in medusozoan and anthozoan cnidarians, we detect neurons receiving BMP signals, which suggests that pro-neural role of BMP signaling is an ancestral cnidarian trait predating the anthozoan-medusozoan divergence.

### BMP signaling promotes neurogenesis in parts of bilaterian nervous systems

BMP signaling is associated with various stages of neurogenesis in Bilateria, displaying negative or positive regulatory effects depending on the specific location and timing within the developmental context. In vertebrates and arthropods, the specification of the centralized nervous system requires inhibition of BMP signaling, while differing levels of BMP activity regulate the mediolateral patterning of neurectoderm by homeobox factors *nk2.1, nk2.2, nk6, pax6, pax3/7* and *msx* [64]. The question about neurogenic functions of BMP signaling becomes more complicated when we consider animal groups outside of vertebrates and arthropods. Spiralians, ambulacrarians and xenacoelomorphs display an assortment of centralized and decentralized neural architectures that, in many cases, differ in their developmental regulation from what is known from groups with centralized nervous systems.

Little is known about neuronal roles of BMP signaling in xenacoelomorphs, whose phylogenetic position, either at the base of Bilateria or within the deuterostomes, is still unresolved [65–70]. The group displays divergent neural architectures, ranging from diffuse nets with simple nerve plexuses to organized nervous systems with multiple nerve cords [65,71]. BMP signaling inhibition in a xenacoelomorph nemertodermatid did not affect the establishment of the nerve cords but increased the number of serotonin/(5-HT)-positive commissures [16].

In nonchordate deuterostomes, different forms of decentralized nervous systems exist, including the pentaradial nerve cords and nerve ring in echinoderms and the epidermal nerve net in hemichordates, also featuring a ventral and the dorsal nerve cord [72]. Like in chordates, BMP signaling appears to initially repress neurogenesis in ambulacrarians [9], and later, may be required for local neural specification [73]. In the sea urchin, BMP signaling locally promotes the pro-neural markers *soxC, soxB1* and *sip1,* as well as serotonergic neurogenesis only in the anterior neural ectoderm, but not in the ciliary band or in the region of the endomesoderm that later will give rise to the gut [17]. In contrast, BMP signaling disruption in the sea star had no effect on *elav* expression but affected the localization of *elav*-positive cells [74]. In the hemichordate *Saccoglossus*, exogenous BMP had no anti-neurogenic effects, based on the persisting expression of pan-neural marker *elav* [13].

Neural anatomies in spiralians are diverse, often displaying different numbers of nerve chords. In the context of neurogenesis, BMP signaling was found to have positive or negative effects on the formation or organization of nerve chords, the brain or eyes [18,75–80]. In the snail *Ilyanassa*, BMP signaling was shown to promote the neural ectoderm and the formation of eyes and the shell [75], while another gastropod, *Crepidula*, exhibits enlarged cerebral ganglia and ectopic ocelli when BMP signaling is blocked by the selective Activin receptor-like kinase-2 inhibitor Dorsomorphin Homolog 1 (DMH1) [15]. In the annelid worm *Chaetopterus*, BMP signaling inhibition had no effect on neural structures [76]. Similarly, a recent analysis in the early branching annelid *Owenia*, where a BMP signaling gradient is responsible for the DV patterning, showed that modulating BMP signaling also did not prevent the development of neural structures or differentiation of specific neuron types [81]. Until now, no information on the areas of BMP signaling activity was available for the nonlophotrochozoan spiralians. We analyzed BMP signaling activity in the embryos, hatchlings and juveniles of the chaetognath *Spadella* and showed that a dorsal-to-ventral BMP signaling gradient forms during its early development. *Spadella* has a

highly centralized ventral nervous system, the ventral nerve center, which appears to be patterned mediolaterally by the same genes as the central nervous systems of vertebrates, insects and polychaetes [54]. Importantly, the chaetognath ventral nerve center forms from two bilateral patches of neurectoderm located in the lateral sides of the ectoderm of the embryo. These regions give rise to multiple neural components, including the lateral somata clusters, which contribute to the developing nervous system. Here we show that during *Spadella* development, dorsal parts of the neurogenic ectoderm are specified within the active BMP signaling domain in late gastrula. We detect BMP signaling in the dorsal parts of the lateral somata clusters (neuroectoderm) of the early elongation stage embryos and, subsequently, in parts of the ventral nerve center and transiently also in some of the sensory organs such as the corona ciliata in the hatchling and in the juvenile.

Chaetognaths are not the only animals with highly centralized nervous systems, where BMP signaling is active in the neural structures—especially in the peripheral and enteric neurons. In the annelid worm *Platynereis* and the fly *Drosophila*, which both have a ventral nerve cord, the expression of pro-neural marker *atonal,* as well as the formation of peripheral sensory neurons were promoted by BMP signaling [2,82]. In the cephalochordate *Amphioxus*, BMP signaling inhibition is required for CNS formation [83], whereas high BMP signaling promotes the formation of *elav*-positive peripheral sensory neurons [14]. Similarly, BMP signaling in the tunicate *Ciona* is involved in the induction and differentiation of the peripheral nervous system [84,85]. In several vertebrate models (fish, chick, mouse and human), BMP signaling is associated with promoting the enteric nervous system [86,87]. In the zebrafish, pSMAD1/5 activity overlaps with *phox2b*-positive enteric progenitor cells and *elav*-positive differentiated enteric neurons, regulating either the generation of progenitor cells or their differentiation at different time points of embryonic and larval development [87]. But the role of BMP signaling in vertebrate nervous system development is not limited to the peripheral neurons. A ventral-to-dorsal Sonic hedgehog signaling gradient and a dorsal-to-ventral BMP signaling gradient in the vertebrate neural tube are responsible for its DV patterning [88]. However, the dorsalmost neural tube cells, in which BMP signaling is strongest, descend from the ventralmost neuroectodermal cells, which are closest to the BMP signaling maximum. Therefore, BMP-dependent patterning of the vertebrate neural tube appears to be not a separate, "late" role of BMP signaling kicking in once its initial anti-neural function has been fulfilled, but rather a continuous BMP-dependent DV patterning of the whole ectoderm, including the neuroectoderm (S6 Fig). Taken together, rather than being an unusual feature of cnidarians with their simple, diffuse nervous systems, pro-neural BMP signaling activity is also observed across Bilateria where it promotes neurogenesis of peripheral, sensory and enteric neurons and plays a part in the chordate CNS patterning.

## Conclusions and outlook

Our analyses suggest that positive neuronal regulation by BMP signaling may represent an ancestral feature, predating the cnidarian-bilaterian divergence. We propose that the "anti-neural" role of BMP signaling documented in vertebrates and arthropods (Fig 11D) is a consequence of the global program of the BMP-dependent DV patterning of the ectoderm in the animals where neurectoderm is represented by a contiguous domain at the "low BMP signaling" side of the DV axis. In the future, it will be important to analyze the expression and the functions of the transcription factors responsible for the mediolateral patterning of the central nervous system in bilaterians with multiple nerve cords as well as in the ones with diffuse nervous systems. If we find evidence of involvement of the mediolateral patterning transcription factors in global patterning of the ectoderm, this will be a strong argument in favor of the "simple urbilaterian" hypothesis [11,12] and multiple independent centralizations of the nervous system across the bilaterian tree. In contrast, evidence of mediolateral TFs specifically controlling neuronal differentiation rather than general ectodermal patterning in the models with decentralized nervous systems would suggest that a high degree of nervous system centralization was already present in the "urbilaterian" [4,8] and was later independently lost in multiple bilaterian lineages.

## Materials and methods

### Animal culture

Adult polyps of *Nematostella vectensis* were maintained in the dark at 18 °C in 16 ppt artificial seawater (*Nematostella* medium, NM) and spawned as described before [89,90]. *Tripedalia* medusae were collected at the aquarium of the Tiergarten Schönbrunn in Vienna and fixed and processed for antibody staining on the same day. *Aurelia* polyps were kept in Petri dishes in artificial sea water (ASW, 35‰) at 20 °C and fed with *Artemia* once a week. Strobilation was induced by keeping the polyps at 15 °C overnight or by incubation in 20 µM Indomethacin/ASW for 3 days. Ephyra were transferred to a small beaker with ASW and fed with *Artemia* once a week. *Stomolophus sp.* metaephyrae were provided by the "Haus des Meeres" aquarium (Vienna, Austria), and maintained as described for *Aurelia*. Live *Spadella cephaloptera* specimens were collected from the intertidal zone near Roscoff, France, and maintained under laboratory conditions until spawning, following established procedures [54,91].

### Single-cell RNA-seq analysis

Analysis of single-cell transcriptomic data was performed using previously published data [37]. Gene expression of genes of interest in the developmental dataset and the neuroglandular subset was examined using the Seurat::DotPlot function with maximum dot size fixed at 100%. R scripts used for the processing and analysis of the *Nematostella* single-cell data are available on GitHub at https://github.com/Genikhovich-Lab/Data-for-plots/. The R-object containing all the raw data used in this study is available at https://cells.ucsc.edu/sea-anemone-atlas/Nv2/all/AllData.Robj [37]. For each gene and cell group (ID.separate), all individual cell-level expression values can be obtained using the Seurat package by running an R-script available at https://github.com/Genikhovich-Lab/Data-for-plots/blob/main/scRNA-SeqDotPlots.R. Summary statistics used for visualization in dot plots were calculated as follows: mean expression represents the average normalized expression across all cells within a group; the percentage of expressing cells represents the fraction of cells with nonzero expression. Standard deviation (SD) and standard error of the mean (SEM) were calculated across cells within each group. No additional normalization or filtering beyond Seurat defaults was applied. Additional data processing steps required to analyze the expression of our genes of interest in stem cells are described in the R-script https://github.com/Genikhovich-Lab/Data-for-plots/blob/main/scRNA-seqDotplots_Fig2_FigS4.R. The content of the https://github.com/Genikhovich-Lab/Data-for-plots/ is additionally archived at https://doi.org/10.5281/zenodo.19686897.

### Immunostaining and vibratome sectioning

Antibody staining in *Nematostella* polyps, *Stomolophus* metaephyra, *Aurelia* ephyra and *Tripedalia* medusa was performed as described previously [36]. For co-staining pSMAD1/5 and neuronal markers, *Aurelia* and *Stomolophus* were fixed for 2 min in cold 0.25% glutaraldehyde/3.7% formaldehyde in PTx (1x PBX, 0.2% Triton-X100), followed by 1 h fixation in 3.7% formaldehyde in PTx at 4 °C. After several washes with PTx, samples were incubated in Blocking solution (1% BSA, 5% sheep serum, 1 × PBS, 0.2% Triton-X100, 20% DMSO) at room temperature for at least 2 h. The primary antibodies (1:200 rabbit anti-pSMAD1/5/9 (Cell signaling, #13820), 1:500 rabbit anti-GLWamide 1675 III p [49], 1:1000 mouse anti-beta-Tubulin (Proteintech, 66240-1-Ig), 1:500 rabbit anti-Serotonin/5-HT (Immunostar, CAT20080), 1:500 rabbit anti-FMRF (Neuromics, RA20002)) were pre-absorbed in Blocking solution (1% BSA, 5% sheep serum, 1 × PBS, 0.2% Triton-X100, 0.1% DMSO) and incubated with the sample at 4 °C overnight. Samples were washed 10 times in 1 × PBS/0.2% Triton-X100, incubated in Blocking solution at room temperature for 1 h and stained with the secondary antibodies at room temperature for 2 h or at 4 °C overnight. Secondary antibodies were diluted 1:1000 in Blocking solution (goat α-rabbit IgG-Alexa633 (Invitrogen A21070), goat α-mouse IgG-Alexa488 (Invitrogen A11001), goat α-mouse IgG-Alexa568 (Invitrogen A11004) and 5 µg/ml DAPI) Samples were washed 10 times in 1 × PBS/ 0.2% Triton-X100, and either infiltrated with and mounted in Vectashield Antifade Mounting Medium (H-1000–10, VectorLabs) or processed for vibratome

sectioning. For the sectioning, samples were embedded in 10% gelatin/PBS, fixed in 4% Formaldehyde in PBS at 4 °C overnight and sectioned at a Leica VT1200 vibratome as described before. All stainings were performed at least two times independently with three or more animals imaged. Images processing and figure preparation were performed using Fiji [92] and Affinity Designer (https://affinity.serif.com/en-us/designer/).

*Spadella* embryos and post-hatching stages were fixed in 4% paraformaldehyde in MOPS buffer (0.1 M MOPS pH 7.4, 2 mM EGTA, 1 mM MgSO$_4$, 2.5M NaCl). For pSMAD1/5 immunostaining, samples were stored in 70% ethanol at −20 °C. pSMAD1/5 immunostaining was performed on early and late gastrulae, early elongation stage individuals, hatchlings, and early juveniles (5 days post-hatching). Samples were rehydrated through an ethanol/PBS series (50%, 25%, PBS) and washed three times in PBS-TX (0.3% Triton X-100 in PBS). Blocking was carried out overnight at 4 °C in PBS-TX with 3% normal goat serum (NGS), followed by overnight incubation with anti-pSMAD1/5/9 antibody (1:100; Cell Signaling #13820) in PBS-TX + 3% NGS. After a series of washes in PBS-TX for 2 hours, specimens were incubated overnight at 4 °C with goat anti-rabbit Alexa Fluor 488 (1:1000; Invitrogen #A-11094) and DAPI (1 µg/ml) in PBS-TX + 1% NGS. After a final series of washes, samples were mounted in Vectashield Antifade Mounting Medium (Vector Laboratories).

## Morpholino knockdowns, inhibitor treatments and in situ hybridization in embryos

For knockdowns of *bmp2/4* and *gdf5-like* in *Nematostella*, previously published morpholino oligos were injected into fertilized eggs as described before [33,35]. For chemical BMP signaling inhibition, embryos were incubated in *Nematostella* Medium (NM) with 6 µM K02288 or, as control treatment, with equal volumes of DMSO. Treatments were carried out starting from blastula stage (18 h post fertilization, hpf). For the wash-out at 2-day (48 hpf), 4-day (96 hpf) or 5-day (120 hpf) planula stage, embryos were washed three times with NM. Morpholino and K02288 treated embryos were fixed for in situ hybridization as described before [30,93] with minor modifications. *Nematostella* planulae were fixed in 4% PFA in 1 × PBS and 0.1% Tween 20 (PTW) for 1 hr at room temperature. For permeabilization, samples were incubated in 10 µg/ml Proteinase K/PTW for 20 min at room temperature. Following the 2 × SSC wash, 4 d planulae were incubated in 1 unit/µl RNAseT1 in 2 × SSC at 37 °C for 40 min and then washed with 0.075 × SSC to reduce unspecific "collar" staining around the pharynx.

*Spadella* embryos used for HCR-FISH with *bmp2/4* and *chordin* probes were fixed at gastrula stage as described above and stored in 100% methanol at −20 °C (rather than in 70% ethanol). HCR probes were designed with the insitu_probe_generator.py script (https://github.com/rwnull/insitu_probe_generator; [94] and synthesized by Integrated DNA Technologies. Hairpins were purchased from Molecular Instruments (USA). HCR-FISH followed the protocol of Choi *et al*. [95], with additional modifications from Bruce *et al*.[96]. Cell nuclei were counterstained with DAPI during the final incubation step. Samples were then incubated in Vectashield for 20 min, cleared in a graded TDE (2,2′-thiodiethanol) series in PBS (30%, 60%, and 80%), and mounted in 80% TDE for imaging. Imaging was performed with a Leica TCS SP5 confocal microscope (Leica Microsystems, Germany). Image processing and figure preparation were carried out using Fiji [92] and Inkscape (https://inkscape.org).

## Inhibitor treatments and bulk RNA-seq of adult *Nematostella* polyps

Adult *Nematostella* polyps were incubated in 5mM lidocaine in NM for 5 min or until fully relaxed and transferred to a 6-well plate containing 20 µM K02288 in NM or same volume of DMSO in NM as control. For a more even exposure, the solution was injected into the gastrovascular cavity through the mouth using a 1 ml syringe and a blunt Sterican needle with a bent tip. Animals were treated for 5 h, relaxed for 10 min using 5mM lidocaine in treatment solution and then transferred to 100% methanol for dissection. Total RNA extraction was performed using the Zymo Research Quick-RNA Miniprep Kit Plus, following the manufacturer's instructions for sample preparations of tissues.

Total RNA was extracted from the female mesenteries, male mesenteries, female body wall and female head of the K02288 and DMSO-treated polyps in biological triplicates and shipped to Novogene for polyA-enrichment, library

preparation and paired-end 150 bp Illumina sequencing. Adapters were trimmed using Cutadapt 4.4 [97], and the reads were aligned to the *Nematostella vectensis* genome [98] using star/2.7.10b [99]. Expression values were obtained using featureCounts in Subread/2.1.1 [100] and differential expression analysis was performed with DeSeq2 [101]. For the analysis, expression changes with an adjusted *p*-value of <0.05 were used. Datasets generated in this study were compared with the previously published anti-pSMAD1/5 ChIP dataset (late gastrula and 4-day planula stage) and the BMP2/4 morpholino (BMP2/4MO) RNA-seq data (2-day planula) [30]. The formerly used NVE gene models were matched to the new NV2 gene annotations. NVE gene models with no clear NV2 ID were not considered for further analysis. NVE-to-NV2 matching reduced the number of gene models used for the comparison, e.g., of 254 NVE models identified as direct targets of BMP signaling by ChIP-Seq [30], 210 had a one-to-one NV2 counterpart.

**Inhibitor treatments, bulk RNA-seq, de novo transcriptome assembly and differential gene expression analysis in *Stomolophus***

*Stomolophus* metaephyrae were treated for 16 h with DMSO or 6 µM K02288 and total RNA was extracted from 4 biological replicates for each treatment. The samples were shipped to Novogene, where nondirectional Illumina sequencing libraries were prepared from polyA-selected samples and paired-end 150 bp Illumina sequencing was performed. Trimmomatic v0.40 [102] was used to remove adapter leftovers and low quality stretches using a sliding window of 4 bp and a minimum average quality of 20 and a minimum read length after clipping of 75 bp. Trimmed reads were then assembled using Trinity v2.15.2 [103] using default parameters. Orthologous sequences between *Nematostella,* human and *Stomolophus* transcriptomes were identified with Orthofinder v3.1.1 using Diamond as the default search algorithm, famsa for the multiple sequence alignment and fasttree for fast phylogenetic tree construction. Pseudomapping of the reads was performed using Kallisto v0.52.0 [104] with default parameters. Differential gene expression analysis was performed in R using tximport v 1.34.0 [105] for importing and consolidating read counts and DESeq2 v1.46.0 [101] for differential gene expression. R scripts for reproduction of the analysis are available in GitHub https://github.com/Genikhovich-Lab/Stomolophus and additionally archived at https://doi.org/10.5281/zenodo.19686882.

## Supporting information

**S1 Fig. BMP signaling is detectable in nematocyst batteries of the *Tripedalia* exumbrella.** ncb - nematocyst battery, Scale bars 50µm.
(TIFF)

**S2 Fig. GO-enrichment analysis of the differentially expressed genes upon K02288 treatment of adult polyp tissues.** Numerical data for the bar plot can be found in the K02288_adj.GOenrich sheet of the S1 Table.
(TIFF)

**S3 Fig. *bmp2/4* is expressed dorsally and *chordin*—ventrally in the midgastrula of the chaetognath *Spadella*.** (A-A') Lateral view and (B-B') frontal view and the corresponding sketches of the anatomy of the embryo. Orange outlines demarcate the neuroectoderm (A, B) and dashed outlines demarcate the endoderm (B). Scale bars 50 µm. Abbreviations: ect, ectoderm; eme, endomesoderm; nec, neuroectoderm.
(TIFF)

**S4 Fig. The expression of BMP signaling components in the developmental subset of primordial stem cells and neuronal progenitors is enriched in the *soxB2*-positive subcluster pSC.NPC.1.** Average scaled expression of 0 or below is indicated in gray. Numerical data for the plot can be extracted by running the R scripts found here https://doi.org/10.5281/zenodo.19686897.
(TIFF)

**S5 Fig. BMP signaling activity in the ephyra of the moon jellyfish *Aurelia* is present in the muscle field of the central disc, cnidocytes and parts of the rhopalia.** (A) Life cycle schematic of *Aurelia coerulea*, details of the ephyra juvenile stage. (B) Rhopalia in the lappets exhibit pSMAD1/5 staining at the rhopalic canal (rc). (C) pSMAD1/5 staining in the central disc of the ephyra. Pha - phalloidin. (C') Detail showing pSMAD1/5-positive crescent-shaped nuclei typical for cnidocytes. cd - central disc, rh - rhopalium, rc - rhopalial canal, bs-basal segment, in-intermediate segment, te-terminal segment; Scale bar (B-B') 25 μm, (C) 50 μm, (C') 5 μm.
(TIFF)

**S6 Fig. Scheme of the BMP signaling-dependent patterning of the vertebrate ectoderm in case of primary neurulation.** The dorsalmost part of the neural tube originates from the ventralmost neurectoderm. Blue circles indicate pSMAD1/5 positive nuclei.
(TIFF)

**S1 Plot. Dot plot sorted by similar expression of *Nematostella* orthologs of differentially expressed *Stomolophus* transcripts with differential expression in K02288 treated adult *Nematostella* polyps.** The R-object containing all the raw scRNA-Seq data used in this study is available at https://cells.ucsc.edu/sea-anemone-atlas/Nv2/all/AllData.Robj. The list of gene models and the R script "scRNA-SeqDotPlots_S1_Plot-S4_Plot.R" allowing to reproduce this plot and extract the underlying data can be found at https://doi.org/10.5281/zenodo.19686897.
(PDF)

**S2 Plot. Dot plot of *Nematostella* orthologs of differentially expressed *Stomolophus* transcripts with differential expression in BMP2/4 2d *Nematostella* morphants, specifically expressed in neurons.** The R-object containing all the raw scRNA-Seq data used in this study is available at https://cells.ucsc.edu/sea-anemone-atlas/Nv2/all/AllData.Robj. The list of gene models and the R script "scRNA-SeqDotPlots_S1_Plot-S4_Plot.R" allowing to reproduce this plot and extract the underlying data can be found at https://doi.org/10.5281/zenodo.19686897).
(PDF)

**S3 Plot. Dot plot of *Nematostella* orthologs of differentially expressed *Stomolophus* transcripts with differential expression in BMP2/4 2d *Nematostella* morphants, specifically expressed in cnidocytes.** The R-object containing all the raw scRNA-Seq data used in this study is available at https://cells.ucsc.edu/sea-anemone-atlas/Nv2/all/AllData.Robj. The list of gene models and the R script "scRNA-SeqDotPlots_S1_Plot-S4_Plot.R" allowing to reproduce this plot and extract the underlying data can be found at https://doi.org/10.5281/zenodo.19686897).
(PDF)

**S4 Plot. Dot plot sorted by similar expression of *Nematostella* orthologs of differentially expressed *Stomolophus* transcripts encoding TF with differential expression in BMP2/4 2d *Nematostella* morphants.** The R-object containing all the raw scRNA-Seq data used in this study is available at https://cells.ucsc.edu/sea-anemone-atlas/Nv2/all/AllData.Robj. The list of gene models and the R script "scRNA-SeqDotPlots_S1_Plot-S4_Plot.R" allowing to reproduce this plot and extract the underlying data can be found at https://doi.org/10.5281/zenodo.19686897).
(PDF)

**S1 Table. List of differentially expressed genes upon K02288 treatments of adult polyps.**
(XLSX)

**S2 Table. Comparison of K02288 RNA-Seq, BMP2/4MObmp2/4MO RNA-Seq and LG4dP anti-pSMAD1/5 ChIP-Seq datasets.**
(XLSX)

**S3 Table. List of *Stomolophus* differentially expressed transcripts upon K02288 treatment (padj < 0.05), *Nematostella* gene models orthologous to the differentially expressed *Stomolophus* transcripts and comparison of the differential expression in Stomolophus with the DEG lists of K02288 treatments of adult *Nematostella* polyps and 2dpf BMP2/4MO-injected *Nematostella* embryos.**
(XLSX)

## Acknowledgments

Confocal microscopy was performed at the Core Facility Cell Imaging and Ultrastructure Research, University of Vienna - member of the Vienna Life-Science Instruments (VLSI). Computational work was performed on the Life Science Compute Cluster (LiSC) of the University of Vienna. We thank Fabian Rentzsch (University of Bergen) for providing *soxb(2)::mOrange* and *prdm14d::gfp* transgenic lines of *Nematostella*, Anders Garm (University of Copenhagen) and Yulia Kraus (Lomonosov Moscow State University) for help with interpreting the morphology of the *Tripedalia* rhopalium, and Sanjay Narayanaswami for the discussions.

## Author contributions

**Conceptualization:** Paul Knabl, Tim Wollesen, Grigory Genikhovich.

**Data curation:** Juan Daniel Montenegro Cabrera.

**Formal analysis:** Paul Knabl, Juan Daniel Montenegro Cabrera, Grigory Genikhovich.

**Funding acquisition:** Paul Knabl, Tim Wollesen, Grigory Genikhovich.

**Investigation:** Paul Knabl, June F. Ordoñez, Tim Wollesen.

**Methodology:** Paul Knabl, June F. Ordoñez, Juan Daniel Montenegro Cabrera, Grigory Genikhovich.

**Project administration:** Grigory Genikhovich.

**Resources:** Daniel Abed-Navandi, Roland Halbauer, Oliver Link.

**Supervision:** Tim Wollesen, Grigory Genikhovich.

**Visualization:** Paul Knabl, June F. Ordoñez.

**Writing – original draft:** Paul Knabl, Grigory Genikhovich.

**Writing – review & editing:** Paul Knabl, June F. Ordoñez, Tim Wollesen, Grigory Genikhovich.

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
