## [Editor Report · Decision Letter 0]

16 Jun 2025

Dear Dr Genikhovich,

Thank you for submitting your manuscript entitled "The anti-neural role of BMP signaling is a side effect of its global function in dorsoventral patterning" for consideration as a Research Article by PLOS Biology.

Your manuscript has now been evaluated by the PLOS Biology editorial staff as well as by an academic editor with relevant expertise and I am writing to let you know that we would like to send your submission out for external peer review.

Once your full submission is complete, your paper will undergo a series of checks in preparation for peer review. After your manuscript has passed the checks it will be sent out for review. To provide the metadata for your submission, please Login to Editorial Manager (https://www.editorialmanager.com/pbiology) within two working days, i.e. by Jun 18 2025 11:59PM.

Kind regards,

Ines

--

Ines Alvarez-Garcia, PhD

Senior Editor

PLOS Biology

---

## [Decision Letter · Decision Letter 1]

30 Sep 2025

Dear Dr Genikhovich,

Thank you for your patience while your manuscript entitled "The anti-neural role of BMP signaling is a side effect of its global function in dorsoventral patterning" was peer-reviewed at PLOS Biology. Please also accept my sincere apologies again for the long delay in sending you our decision. The manuscript has now been evaluated by the PLOS Biology editors, an Academic Editor with relevant expertise, and by two independent reviewers.

The reviews are attached below. As you will see, both reviewers find the conclusions interesting, but they also raise several issues that would need to be addressed before we can consider the manuscript for publication. Reviewer 1 thinks that the findings would be more convincing if the pSMAD1/5 staining results in medusozoan and Chaetognath would be also supported by functional studies and makes some suggestions to achieve this. In addition, this reviewer asks for several clarifications and offers an alternative explanation to support the hypothesis on the multiple independent centralisations of the nervous system across the bilaterian tree that should be considered. Reviewer 2 is very positive, but thinks that some of the wording included in the title and abstract should be improved, and asks for several clarifications in the text.

Given the extent of revision needed, we cannot make a decision about publication until we have seen the revised manuscript and your response to the reviewers' comments. Your revised manuscript is likely to be sent for further evaluation by all or a subset of the reviewers.

**IMPORTANT - SUBMITTING YOUR REVISION**

3. Resubmission Checklist

a) *PLOS Data Policy*

b) *Published Peer Review*

Sincerely,

Ines

--

Ines Alvarez-Garcia, PhD

Senior Editor

PLOS Biology

Reviewers' comments

Rev. 1:

In this manuscript, the authors show that BMP signaling is active in the adult sea anemone Nematostella within particular neuronal cell types. Through functional experiments using BMP receptor inhibitor and RNA-seq analysis, they also demonstrate that BMP signaling can positively regulates neuronal gene expression and neurogenesis. They also show that BMP signaling is active in the medusozoan nervous system, suggesting it may function in promoting neurogenesis for both anthozoans and medusozoans. Finally, they show that BMP signaling is also active in the neurogenic domain of Chaetognath embryos, and later in the sensory organ of the juvenile. Based on these results, the authors suggested that BMP signaling can positively regulate neurogenesis in a wide variety of animals and may represent an ancestral feature, predating the cnidarian-bilaterian divergence. They also propose that the "anti-neural" function of BMP signaling in classical bilaterian models may represent the co-option of its global role in the dorsoventral patterning of the ectoderm.

In general, this manuscript is well written, and the data presented are mostly well presented and robust, especially for the parts on Nematostella. However, I do have some comments regarding the observation of BMP activity in medusozoan and Chaetognath, as well as on its activity in relation to neurogenesis in these animals. More specifically, the weakness of this manuscript is that the authors did not provide experimental evidence to support the "pro-neural" function of BMP signaling in medusozoan and Chaetognath. Therefore, while the results of pSMAD1/5 staining in these animals are quite striking, there remains a large degree of uncertainty about the conclusions of BMP activity for their neural development. It would become much more convincing if the pSMAD1/5 staining results in medusozoan and Chaetognath were also integrated by functional studies, such as the experiments applying BMP receptor inhibitor K02288 (like that conducted in Nematostella) or recombinant BMP protein to elevate BMP signaling level (see Denes et al., Cell 2007, 129:277-88).

About the Aurelia ephyra data presented in Fig 8 and the text on page 17, it is uncertain to me whether the observed pSMAD1/5 signals in the rhopalic canal are really located in neuronal cells. Although the authors mentioned that they were unable to combine the staining of neuronal markers with pSMAD1/5 staining, why not using antibodies against neuropeptides (for example, see Nakanishi et al., Dev Genes Evol 2009, 219:301-17. doi: 10.1007/s00427-009-0291-y) to label neuronal cells in comparable samples separately and then compare the result with pSMAD1/5 staining to see whether they are showing similar pattern?

It is interesting to see that the dorsal expression of BMP2/4 and the ventral expression of Chordin seems to be consistent with the observed graded pSMAD1/5 signals in the developing Chaetognath embryo (Fig 9 and S1 File-Fig 2). It is not mentioned whether this early BMP signaling gradient might be involved in the specification and patterning of neuroectoderm in the ventral part of the Chaetognath gastrula. More specifically, is inhibition of BMP signaling necessary for the initial specification of ventral nervous system in Chaetognath embryo? This is an important question that needs to be address in the manuscript.

In the last part of the Discussion (page 25), the authors discuss about the evolutionary scenario of "multiple independent centralizations of the nervous system across the bilaterian tree" and the necessary evidence to support this hypothesis. However, this hypothesis seems to be based on the assumption that diffused nervous system represents the ancestral condition in both protostome and deuterostome lineage, and that separate co-option of BMP-dependent DV patterning mechanism for the initial specification of neural ectoderm occurred multiple times for the evolution of CNS in different lineages. However, it seems equally possible that this co-option happened once in the common ancestor of bilaterians, and subsequently modified or lost in some lineages, resulting in a less centralized nervous system in some animals. I am curious how the authors could exclude this "single origin" scenario.

Other minor points:

1. Fig 1 and its legend; please note that in 1C, it was not mentioned explicitly the red line indicates the directive axis and the blue dots indicate pSMAD1/5 activity; please confirm my interpretation. In addition, many abbreviations in 1D and 1E are not explained in the legend.

2. Page 11, the sentence "Among the different tissues only 9 genes were differentially regulated in all tissue samples (Fig 5C, S1 Table)"; This citation of figure should be "Fig 5D', and in Supplementary Table S1, I only saw 8 gene listed on the page "ov_all_tissues" and the number indicating "#DEGs present in all tissues" is also listed as 8. Please confirm this number.

3. Page 11, the sentence "Seven of these 20 were transcription factors or co-factors (Fig 5D, S2 Table)", the correct figure citation should be "Fig 5E". Please confirm.

Rev. 2:

This manuscript presents a thorough dissection of relationship between BMP2/4 activity and neural cell types and development in the anemone Nematostella. This is a very timely topic because there have been prominent arguments that a role of the BMP2/4 pathway in repressing neurogenesis is ancestral for bilaterians (based on evidence from vertebrates and flies). In many other taxa BMP activity co-occurs with neurogenesis indicating that it does not repress it. Thus, the further examination of this question in Nematostella and a cubomedusan represent an important advance, largely because of the position of cnidarians as an outgroup to Bilateria. The data is generally high quality, and well-presented and the writing is clear. I have only minor comments.

Minor comments

Title, abstract, elsewhere: "side effect" is not quite right; it's vague about something that we have specific terms for. I think what they are arguing is that it is a convergent trait that evolved in two lineages where the nervous system became highly centralized. Even "secondary" or "independently derived" would be more precise. Here is a suggestion for implementing that in the title.

The anti-neural role of BMP signaling in some taxa is independently derived from its ancestral role in global dorsoventral patterning

"contradicted by the variety of the degrees of centralization of the bilaterian nervous systems - from

diffuse to being arranged into a varying number of nerve cords to fully centralized"

I think the "ganglial" level of NS organization is an important category to recognize—many groups have nervous systems where there are multiple large ganglia connected by relatively small numbers of neurons that would not be called a net or a cord.

"While neurons of the rhopalia are not marked by αTub,

many pSMAD1/5-positive cells in the area between the stalk and the upper lens eye are likely to be

neurons based on their location and cell morphology. pSMAD1/5-positive cells in this area exhibit a

high cytoplasm to nucleus ratio (Fig 7J, white arrowheads)"

This is not documented really at all. At the very least they would need to have much higher power images of these cells so the viewer could tell how much cytoplasm there was compared to nuclear volume.

In addition, neural marker staining of the rhopalium would be useful to show that the staining is in a region of neurons.

If they can't get these things, they should rephrase what these data imply, to something like "The location of these pSMAD+ cells is consistent with at least some having neural identities."

Figure 7, panel L, indicates the yellow stain is Pha, but there is no indication of what this is? Phalloidin?

Last figure, panel D: Is there enough known from Knabl 2024, and Watanabe 2014 to include planula and or embryo stages of Nematostella in this panel along with polyp? These would seem to be more apt comparisons to the embryonic events summarized for other taxa here.

Also this panel, it's unclear to me how to understand the "mediolateral" patterning diagrams.. could they be put on a circular cross-section or is that not what they are trying to convey?

Also, on the top row of that panel, maybe put the "elav+" at the beginning (Danio) and/instead of the end?

---

## [Decision Letter · Decision Letter 2]

17 Apr 2026

Dear Dr Genikhovich,

Thank you for your patience while we considered your revised manuscript entitled "The anti-neural role of BMP signaling is a consequence of its ancestral function in dorsoventral patterning" for publication as a Research Article at PLOS Biology. This revised version of your manuscript has been evaluated by the PLOS Biology editors, the Academic Editor and one of the original reviewers.

Based on the review (attached below), we are likely to accept this manuscript for publication, provided you satisfactorily address the remaining points raised by Reviewer 1. Please also make sure to address the data and other policy-related requests stated below my signature.

We expect to receive your revised manuscript within two weeks.

*Published Peer Review History*

*Press*

Sincerely,

Ines

--

Ines Alvarez-Garcia, PhD

Senior Editor

PLOS Biology

Fig. 2A, B; Fig. 7A; Fig. 8C; Fig. S2; Fig. S4 and Plot S1-4

Please also ensure that figure legends in your manuscript include information on WHERE THE UNDERLYING DATA CAN BE FOUND. For example, you can add a sentence at the end of each corresponding figure legend saying: "The data underlying the graphs can be found in S1 Data." Please also ensure your supplemental data file/s has a legend.

CODE POLICY

Per journal policy, if you have generated any custom code during the course of this investigation, please make it available without restrictions. Please ensure that the code is sufficiently well documented and reusable, and that your Data Statement in the Editorial Manager submission system accurately describes where your code can be found. More information on our Code Policy, what and how to share can be found here: https://journals.plos.org/plosbiology/s/code-availability

Reviewers' comments

Rev. 1:

In this revised manuscript, the authors have added a substantial amount of new data and images to address my previous comments. I have only a few minor points remaining that need to be corrected or clarified by the authors:

1. Page 11, Figure 4 legend; (K) should be (G-I). Please confirm.

2. Page 14, third line; (ts) should be (te). Please confirm.

3. Page 14, 6th line, and Fig.6E; cannot see the "hollow yellow arrowhead" described here.

4. Page 14, 9th sentence (Fig. 6G-G", yellow arrowhead) should be (Fig. 6G-G", black outlined white arrowheads).

5. Page 23, 3rd line, legend of Figure 10; cannot see the "red arrowheads" on Fig. C or C'. Please confirm.

---

## [Editor Report · Decision Letter 3]

4 May 2026

Dear Dr Genikhovich,

Thank you for the submission of your revised Research Article entitled "The anti-neural role of BMP signaling is a consequence of its ancestral function in dorsoventral patterning" for publication in PLOS Biology. On behalf of my colleagues and the Academic Editor, Marianne Bronner, I am delighted to let you know that we can in principle accept your manuscript for publication, provided you address any remaining formatting and reporting issues. These will be detailed in an email you should receive within 2-3 business days from our colleagues in the journal operations team; no action is required from you until then. Please note that we will not be able to formally accept your manuscript and schedule it for publication until you have completed any requested changes.

PRESS

Sincerely,

Ines

--

Ines Alvarez-Garcia, PhD

Senior Editor

PLOS Biology
